# Large Language Models Assume People are More Rational than We Really are

**Ryan Liu\***
Department of Computer Science
Princeton University
ryanliu@princeton.edu

**Jiayi Geng\***
Department of Computer Science
Princeton University
jiayig@princeton.edu

**Joshua C. Peterson**
Computing & Data Sciences
Boston University

**Ilia Sucholutsky**
Center for Data Science
New York University

**Thomas L. Griffiths**
Department of Psychology
Department of Computer Science
Princeton University

## Abstract

In order for AI systems to communicate effectively with people, they must understand how we make decisions. However, people's decisions are not always rational, so the implicit internal models of human decision-making in Large Language Models (LLMs) must account for this. Previous empirical evidence seems to suggest that these implicit models are accurate — LLMs offer believable proxies of human behavior, acting how we expect humans would in everyday interactions. However, by comparing LLM behavior and predictions to a large dataset of human decisions, we find that this is actually not the case: when both simulating and predicting people's choices, a suite of cutting-edge LLMs (GPT-4o & 4-Turbo, Llama-3-8B & 70B, Claude 3 Opus) assume that people are more rational than we really are. Specifically, these models deviate from human behavior and align more closely with a classic model of rational choice — expected value theory. Interestingly, people also tend to assume that other people are rational when interpreting their behavior. As a consequence, when we compare the inferences that LLMs and people draw from the decisions of others using another psychological dataset, we find that these inferences are highly correlated. Thus, the implicit decision-making models of LLMs appear to be aligned with the human expectation that other people will act rationally, rather than with how people actually act.

## 1 Introduction

Every day, our actions are based on decisions that reflect our internal goals and beliefs about the world. Through countless interactions with others, we are able to effortlessly predict how other people would act from their goals and beliefs, and infer others' goals and beliefs when observing their actions. These abilities — termed forward- and inverse-modeling in cognitive science (Ho and Griffiths, 2021) — are characteristic of the implicit mental decision-making models that we form of others, and are crucial to interpersonal communication and learning (Baker et al., 2009; Lucas et al., 2014; Jara-Ettinger et al., 2020). However, while these abilities are inherent in people, the consistency and accuracy of decision-making models in Large Language Models (LLMs) is unknown. As LLMs become widely used as the basis for building AI agents that interact with or even simulate people, it is important to ask what decision-making models LLMs implicitly use: the ability to predict and interpret people's behavior is a precursor to identifying effective ways to provide assistance, simulating the helpfulness or harmlessness of a response, and learning individuals' values and preferences, all of which are principal to the development of safe and beneficial AI systems.

Though LLMs have become increasingly capable of conducting reasoning and conversing with humans, there is no guarantee that their implicit representations of humans align with how we behave. Methods such as Proximal Policy Optimization (Schulman et al., 2017) and Direct Preference Optimization (Rafailov et al., 2023) can be used to tune models on explicitly declared human

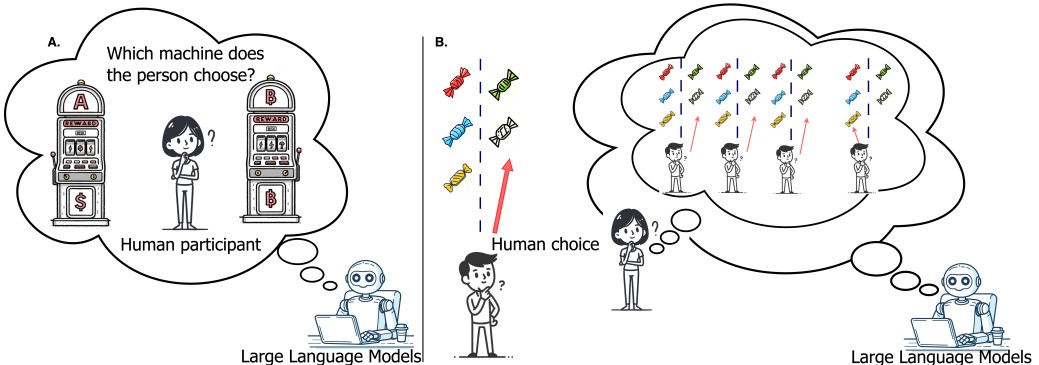

Figure 1: Two tasks we use to assess the implicit assumptions that LLMs make about human decision-making. (A) Predicting choices between gambles. Each gamble is described by the probabilities and values of different outcomes, and the goal is to predict what people will choose. (B) Inferring preferences from choices. Here, a person chooses one of many sets of objects and the goal is to infer their preferences based on that choice.

preferences (Ouyang et al., 2022), but training data such as blog posts, news articles, and books go through rounds of editing that remove logical fallacies and mistakes, leading to more "perfect" content being used for training (Cui et al., 2023; Zhou et al., 2024a). While this improves the quality of generation, it may also lead LLMs to develop mistaken impressions of how humans actually behave. In addition, many of our methods comparing LLMs to human behavior rely on human perception to measure similarity (Park et al., 2023; Jones and Bergen, 2023; Jakesch et al., 2023; Hämäläinen et al., 2023), but the psychology literature has shown that people's perceptions of other people are flawed — we expect others to be more rational than they are (e.g., Jern et al., 2017). Thus, AI systems that act rationally can appear human-like to the naked eye, while not actually capturing human behavior.

This phenomenon is not without precedent — early economists embraced the assumption of human rationality (Smith, 1776; Mill, 1836) and built sophisticated models and policies around it (Downs, 1957; Coleman, 1994; Schelling, 1980; Dunleavy, 2014, *inter alia*), before psychologists showed just how systematic and widespread its failures were in accounting for human behavior (Tversky and Kahneman, 1974; Kahneman and Tversky, 1979). As LLM-powered systems become more widely-used, misaligned representations of humans — which can lead to mismatched beliefs and failure to follow instructions (Milli and Dragan, 2020) — pose a toll on downstream applications. But how can we study LLMs' implicit decision-making models without being affected by the human bias to assume others act rationally?

To explore the implicit decision-making models of LLMs, we leverage two experimental paradigms from psychology — a risky choice task where people choose between gambles (Kahneman and Tversky, 1979, Figure 1A), and an inference task where people infer others' subjective utilities after observing their decisions (Jern et al., 2017, Figure 1B). In the former, we compare choices that participants made with LLM simulations and predictions across a large dataset of over 13,000 risky decisions (Bourgin et al., 2019), while in the latter we compare the inferences drawn by LLMs and by humans over a set of 47 observations (Jern et al., 2017). These two paradigms are connected by foundational theoretical models of human decision-making, under which people develop mental models of others' goals, utilities, and decisions, and use them to 1) construct predictions about how others will behave given their beliefs, and 2) infer what people believe based on their decisions (Baker et al., 2009; Lucas et al., 2014; Jara-Ettinger et al., 2020; Ho and Griffiths, 2021).

Through our experiments, we find that LLMs model people as highly rational decision makers. In the forward modeling task, when prompting with chain-of-thought (CoT; Wei et al., 2022), LLMs consistently predict that people act more rationally than we do. For example, GPT-4o produced a Spearman correlation of $0.94$ with the rational model of choosing the maximum expected value, but humans only have a correlation of $0.48$. Asking LLMs to simulate the decision with CoT also yields highly rational outcomes ($0.90$ for GPT-4o) that are only moderately correlated with human behavior.

In both cases, zero-shot prompting generates noisy outcomes only somewhat correlated with either rational models or humans.

In the inverse modeling task, inferences that LLMs made from peoples' choices were also consistent with assuming that humans are rational. Across two contexts, LLMs' inferences positively correlated with predictions from rational models, where correlations increased with model capability and reasoning (0.20 for Llama-3-8B zero-shot; 0.95 for GPT-4o CoT). Interestingly, psychology literature finds that people, despite deviating from rationality in their own choices, make inferences from the behavior of others as if they were rational (Baker et al., 2009; Lucas et al., 2014; Jern et al., 2017; Jara-Ettinger et al., 2020). Thus, the inferences drawn from others' decisions by people and LLMs should actually be similar. We find support for this: Inferences made by LLMs are highly correlated with the same inferences made by people ($\rho = 0.97$ with GPT-4o CoT). Thus, while LLMs are not accurate at simulating or predicting human behavior, LLMs' incorrect assumption that people are more rational aligns with the assumption that people also make when interpreting another's behavior.

The remainder of the paper is organized as follows. In Section 2, we introduce related work across LLMs, alignment, and decision-making in cognitive science. Section 3 and Section 4 describe the forward and inverse modeling experiments and results, detailing our analyses and the relationships between LLMs, humans, and existing rational theories. Finally, in Section 5 we discuss insights including new challenges for simulating humans using LLMs, misspecification of "alignment" in decision-making, and potential implications for training LLMs using different paradigms.

## 2 RELATED WORK

### 2.1 ALIGNING LARGE LANGUAGE MODELS WITH HUMANS

Large Language Models are typically aligned with human preferences through Reinforcement Learning from Human Feedback (RLHF) (Bai et al., 2022; Ouyang et al., 2022). Training with human preference data has been shown to enhance reasoning (Havrilla et al., 2024), and the impressive capacities of resulting models (e.g., Bubeck et al., 2023) have sparked interest across various fields in using them to model (e.g., Binz and Schulz, 2023a; Macmillan-Scott and Musolesi, 2024) and simulate (e.g., Park et al., 2022; Argyle et al., 2023; Liu et al., 2023; Salewski et al., 2024) human behavior. However, these models still exhibit biases and hallucinations (e.g., Jiang et al., 2024; Bai et al., 2024; Anwar et al., 2024), and may not be adept at capturing trade-offs in human behavior (Liu et al., 2024b; Coletta et al., 2024) or situations with information asymmetry (Zhou et al., 2024b).

Methods from cognitive science are increasingly being used to study LLMs (Coda-Forno et al., 2024; Liu et al., 2024b; Binz and Schulz, 2023b; Liu et al., 2024a). A particularly contentious debate is whether LLMs exhibit Theory of Mind, the ability to model others' mental states which may be different than their own (Premack and Woodruff, 1978). Several studies have shown evidence for (Kosinski, 2023) and against (Sap et al., 2022; Ullman, 2023) Theory of Mind, including studies that develop more rigorous evaluation methods via procedural generation (Gandhi et al., 2024) and adversarial examples (Shapira et al., 2024). Our analysis provides a quantitative approach to engaging with this debate, as inverse decision-making can be considered a specific form of Theory of Mind.

### 2.2 FORWARD AND INVERSE MODELS OF HUMAN DECISION-MAKING

One of the most classic and extensively studied problems in decision-making is the risky choice task (Kahneman and Tversky, 1979; Edwards, 1954; Bourgin et al., 2019), where people choose among gambles with different outcome probabilities and payoffs. The rational action is to choose the gamble with the highest expected value, calculated by summing the product of the probabilities and the values of the outcomes. On this task, humans have been described as deviating from rationality in a fourfold pattern: risk seeking for small probability gains, risk averse for small probability losses, risk averse for moderate or large probability gains, and risk seeking for moderate or large probability losses (Kahneman and Tversky, 1979). Other studies have shown that humans tend to act in accordance with bounded rationality, where rationality is traded-off with mental effort, information, and time (Evans et al., 2015; Alanqary et al., 2021).

Peterson et al. (2021) collected the `choices13k` dataset, a large human dataset with over 13,000 risky choice problems, and showed that people's decisions in this setting could be captured by simple

machine learning models. Binz and Schulz (2023a) used this dataset to fine-tune an LLM, achieving similar performance. Chen et al. (2023) built a risky choice dataset and found that GPT-3.5 makes economically rational decisions, which we replicate in task 3 of our forward modeling experiments on the `choices13k` dataset across various LLMs.

People are able to infer an agent's beliefs, desires, and percepts from their actions (Baker et al., 2017; Lucas et al., 2014; Jara-Ettinger et al., 2020; Ho and Griffiths, 2021). These inferences are typically modeled by assuming that people employ a forward model of decision-making — typically a noisy rational model — and use Bayes' rule to invert that model (for more details, see Section 4). The study that we focus on, by Jern et al. (2017), was intended to directly test this assumption. Similar approaches have been used to align AI systems to preferences inferred from observed user interactions (Fränken et al., 2023) or to improve human-AI interaction (e.g., Dragan et al., 2013; Sadigh et al., 2016). More generally, inverse modeling can also be viewed as inductive reasoning from behaviors to utility functions, where LLMs have been shown to be skilled at proposing hypotheses to explain observations, but not at applying these hypotheses to novel examples (Qiu et al., 2024). Lastly, recent work has extended forward and inverse models using LLMs to make inferences from utterances as well as actions (Zhi-Xuan et al., 2024; Ying et al., 2023).

## 3 FORWARD MODELING: PREDICTING WHICH GAMBLE PEOPLE WILL CHOOSE

**Tasks.** Our forward modeling experiments used the risky choice paradigm, one of the most basic and extensively studied problems in psychological decision theory (Kahneman and Tversky, 1979; Edwards, 1954). In this paradigm, participants face a choice between two gambles $A$ and $B$, each with a set of outcomes that differ in their payoffs $\mathbf{R}$ and probabilities $\mathbf{Q}$. For instance, a risky choice problem might ask, "would you rather win \$5 with probability 1 or take a 0.5 probability of winning \$10?". This can be formalized as gamble $A$ having a single outcome with reward $\mathbf{R}_A = [5]$ and probability $\mathbf{Q}_A = [1]$, and gamble $B$ having two potential outcomes with rewards $\mathbf{R}_B = [10, 0]$ and probabilities $\mathbf{Q}_B = [0.5, 0.5]$. Given a risky choice problem, the goal is to find a probability $P(A)$ that is consistent with how likely people would select gamble $A$ over gamble $B$.

To understand how LLMs empirically capture human intent and align with actual human decisions, we designed three forward modeling tasks. First, we asked LLMs to predict the decisions that a human participant would make. Second, we asked LLMs to predict the proportion of participants that would select each option. Third, we instructed LLMs to simulate participants by making the decisions themselves. The second task corresponds directly to the original objective of finding probability $P(A)$ consistent with how likely people would select gamble $A$ over gamble $B$. In the first and third tasks the LLM outputs a binary decision $\{A, B\}$, which is repeated many times to compute an aggregate probability estimate of choosing gamble $A$, which is compared against $P(A)$.

**Human data.** Human choice data came from the `choices13k` dataset (Peterson et al., 2021; Bourgin et al., 2019), a comprehensive collection of 13,006 risky choice problems. Each choice problem was answered by 15 or more participants, and participants answered each choice problem five times. For each problem, the dataset included the proportion of participant answers that selected each option. Our analyses used a subset of 9,831 problems that were not "ambiguous" (where probabilities were not shown) or lacked feedback. We used the data to evaluate the alignment of LLMs with actual human choices in each of the three paradigms.

**Language models.** We evaluated the following open-sourced and closed-sourced models: Llama-3-8B, Llama-3-70B, Claude 3 Opus, GPT-4-Turbo (0125-preview), and GPT-4o. We implemented experiments on the full `choices13k` dataset for zero-shot and CoT prompting across the three different tasks mentioned above for all models besides Claude. For Claude 3 Opus, we only evaluated the first task — predicting what a human participant's choice might be — due to cost limitations.

For the experiments predicting and simulating individual human decisions (tasks 1 & 3) and models Llama-3-8B, Llama-3-70B, and GPT-4-Turbo, we conducted zero-shot experiments with $n$ completions, where $n$ is the number of participants that made the same decision in the `choices13k` dataset (ranging from 15 to 33). Each completion involved predicting or simulating a single human participant's decision, with temperature set to 0.7 to maintain sample diversity. For the corresponding CoT experiments, we observed that responses were completely deterministic at temperature 0.7 on a

random subset of 1000 decisions, and thus prompted for 1 completion with temperature 0.0. For the same reason, we ran the experiments predicting the proportion of humans who would select an option (task 2) for 1 completion for both zero-shot prompting and CoT prompting with temperature 0.0. To validate our findings, we also ran ablations varying temperature, which we report in Appendix E.1.

We adapted a prompt template previously used by Binz and Schulz (2023a, see Appendix A for example prompts). For each choice problem, we first introduced the decision context of the `choices13k` dataset (i.e., the idea of choosing between gambling options) before providing each option's probabilities and associated rewards in dollars. Finally, we asked what the participant(s) would choose (for predicting decisions) or what "you" would choose (when simulating humans). To minimize any positional biases (Wang et al., 2023), we shuffled the order of the options presented in the prompt. We also conducted an ablation where we assigned demographic profiles to the decision maker to show that our results are consistent across prompt variations (see Appendix E.2).

## 3.1 RESULTS

To evaluate whether LLMs are able to predict or simulate human risky choice decisions, we computed correlations and mean-squared errors (MSE) between LLMs' responses and human decisions. Specifically, we compute the correlations across proportions where the LLM predicts that one gamble will be selected over the other. We report both Pearson and Spearman correlation, but they are extremely similar and we discuss them interchangeably. We also compared LLM responses to a classic model of rational choice: choosing the option with the highest expected value.[1] We focus here on the first of our three tasks — predicting individual choices — because results were similar across the three tasks. Results from predicting the proportion of human choices and simulating decisions are in Appendix B.

**LLMs using zero-shot prompts are poor predictors of human choices.** We found that LLMs with zero-shot prompting are not well-aligned with human decisions. Table 1 shows the model predictions of human behavior against real human choices, where GPT-4-Turbo performed best with a correlation of 0.60 and an MSE of 0.13. LLMs are also not well-aligned with rational decision-making. Table 2 shows that the human correlation with the maximum expected value is 0.48, while GPT-4-Turbo and 4o only achieve correlations of 0.41 and 0.28, with MSEs of 0.27 and 0.36 respectively.[2]

**LLMs using zero-shot prompts sometimes ignore probabilities completely.** To determine exactly what LLMs are doing in the zero-shot case, we fit 18 behavioral models to the responses of GPT-4-Turbo to examine its behavior. These included heuristic models (He et al., 2022), where people are thought to use mental shortcuts to make decisions, counterfactual models (He et al., 2022), which appeal to constructs like regret and disappointment, and subjective Expected Utility models, which assume that quantities involved (money and probability) are perceived or treated subjectively. This includes many of the most influential models including Prospect Theory (Kahneman and Tversky, 1979) and the model proposed by (Peterson et al., 2021), MOT.

In Appendix D, we provide the full analysis with these models. The best interpretation we found uses MOT, where the model fit a mixture of two probability weighting functions — one linear (matching humans), and one approximating a flat line. This suggests that the LLM often expected people to be rational, but completely ignores all probabilities (i.e., weighs them equally) in other cases.

**LLMs using CoT assume people are more rational than they are.** We found all LLMs with CoT prompting assume people act rationally, which is not aligned with actual human decisions. As shown in Tables 1 and 2, even Llama3-8B achieves a correlation with maximum expected value of 0.57, while humans only achieve a correlation of 0.48. This correlation rises as model capabilities improve: Llama3-8B obtains a correlation of 0.57 with an MSE of 0.22; Llama3-70B obtains a correlation of 0.80 with an MSE of 0.1; Claude 3 Opus obtains a correlation of 0.76 with an MSE of 0.12, while GPT-4-Turbo (which best predicted people) and GPT-4o obtain correlations of 0.93 and 0.94 with MSEs of 0.03 and 0.02. The same patterns held in the tasks asking for aggregate behavior or for LLMs to act as humans, although the aggregate behavior task had slightly reduced correlations with

---

[1]Here, Pearson and Spearman correlation are identical as maximizing EV results in a binary response.

[2]In the zero-shot case, we ran a single completion for GPT-4o but multiple for GPT-4-Turbo, which likely resulted in GPT-4o being less correlated with human choices due to variance in sampling.

expected value (see Appendix B). In contrast, we also prompted GPT-4o to make the same predictions for monkeys — which were less rational, but actually closer to human choices (see Appendix E.3).

Table 1: Correlation between LLM predictions of human choices and actual human choices.

|  |  | Llama3-8B | Llama3-70B | Claude3 Opus | GPT-4-Turbo | GPT-4o |
|---|---|---|---|---|---|---|
| Zero-shot | Spearman correlation | 0.3797 | 0.5300 | / | 0.6048 | 0.4756 |
|  | Pearson correlation | 0.3830 | 0.5270 | / | 0.5824 | 0.4718 |
|  | MSE | 0.1283 | 0.1142 | / | 0.1369 | 0.1987 |
| CoT | Spearman correlation | 0.4625 | 0.6156 | 0.5755 | 0.6393 | 0.6113 |
|  | Pearson correlation | 0.4611 | 0.6112 | 0.5750 | 0.6326 | 0.6164 |
|  | MSE | 0.1966 | 0.1633 | 0.1713 | 0.1595 | 0.1638 |

Table 2: Correlation between LLM predictions of human choices and the maximum expected value.

|  |  | Llama3-8B | Llama3-70B | Claude3 Opus | GPT-4-Turbo | GPT-4o | Humans |
|---|---|---|---|---|---|---|---|
| Zero-shot | Spearman correlation | 0.1811 | 0.3378 | / | 0.4106 | 0.2843 | 0.4835 |
|  | Pearson correlation | 0.1811 | 0.3378 | / | 0.4106 | 0.2843 | 0.4835 |
|  | MSE | 0.3145 | 0.3378 | / | 0.2686 | 0.3579 | 0.2580 |
| CoT | Spearman correlation | 0.5665 | 0.7957 | 0.7566 | 0.9322 | 0.9444 | 0.4835 |
|  | Pearson correlation | 0.5665 | 0.7957 | 0.7566 | 0.9322 | 0.9444 | 0.4835 |
|  | MSE | 0.2181 | 0.1031 | 0.1228 | 0.0340 | 0.0278 | 0.2580 |

Although all the LLMs we investigated claim to be aligned with human preferences during the training, our empirical results suggest that these LLMs assume humans act more rationally than they actually do, particularly when CoT prompting is used. In these settings, LLMs correlated more highly with maximizing expected value than with human choices, demonstrating a gap between the implicit model of human decision-making assumed by LLMs and actual human behavior.

## 4 INVERSE MODELING: INFERRING PEOPLE'S PREFERENCES FROM CHOICES

While forward modeling allows us to predict someone's decision given established utilities, inverse modeling is the process of inferring someone's utilities via the decisions they make. Because forward and inverse modeling share the same theoretical framework, such inferences provide a complementary setting to evaluate the decision-making models that LLMs ascribe to humans. To formally measure these inferences, we adapt a psychology experiment from Jern et al. (2017).

**Task.** In the experiment, participants observed 47 decisions made by another person (the observee). In each decision, the observee is shown $n$ groups of items $\{g_1, \ldots, g_n\}$ and choose the group $g_p$ which they prefer. Groups are populated with five distinct types of items, denoted $A, B, C, D, X$. For example, a particular decision could consist of two groups, $g_1 = \{X\}$ and $g_2 = \{A, B\}$, with $g_p = g_1$. Since the participant chose $g_1$, they prefer obtaining $X$ over obtaining both $A$ and $B$.

Participants were asked to rank the 47 observed decisions (which all contained item $X$) based on how much the decision suggested the observee has a high preference for $X$. For instance, the decision $(g_1 = \{X\}, g_2 = \{A, B\}, g_p = g_1)$ shows a higher preference for $X$ than the naive decision $(g_1 = \{X\}, g_p = g_1)$. Participants did not know the utilities of items. Thus, they were only able to determine the observee's preferences based on features such as the number of items in each group.

The 47 decisions covered the most commonly distinguishable decision structures across items $A, B, C, D, X$, under the constraint that each item is included at most once per group. For simplicity, only decisions where $X$ is part of the preferred group $g_p$ were used. We provide a ranking of a subset of close decisions in Table 3, and a full list of 47 decisions in Appendix H.

We found that GPT-4 Turbo could not provide a valid output ranking 47 choices at once. Thus, for all LLMs, we obtained rankings by asking the LLM to perform pairwise comparisons across $\binom{47}{2}$ pairs of decisions. Pairwise outputs were limited to {stronger, weaker, tie}, and were aggregated across decisions to form a ranking. Ties were discouraged to capture small differences between decisions.

Table 3: A subset of the 47 decisions that are closely ranked, ordered by average human ranking. $g_p = g_1$ for all rows. Utilities for items are positive.

| shows less of a preference for X | | | |
| --- | --- | --- | --- |
| $g_1 = \{X\},$ | $g_2 = \{A\}$ | | |
| $g_1 = \{X, A, B\},$ | $g_2 = \{A, B, C\},$ | $g_3 = \{A, B, D\}$ | |
| $g_1 = \{X, A\},$ | $g_2 = \{A, B\},$ | $g_3 = \{A, C\},$ | |
| $g_1 = \{X, A\},$ | $g_2 = \{A, B\},$ | $g_3 = \{A, C\},$ | $g_4 = \{A, D\}$ |
| $g_1 = \{X\},$ | $g_2 = \{A\},$ | $g_3 = \{B\}$ | |
| shows more of a preference for X | | | |

**Dataset.** Observers did not know anything about the values of individual items — instead, relative utilities were inferred after observing the decision maker's choice. Thus, in this paradigm, the remaining four items are equivalently exchangable; choosing {target item $X$} over {item $A$, item $B$} should yield same same inferred level of preference as choosing {target item $X$} over {item $C$, item $D$}. The 47 decisions were structurally unique, yielding coverage over all major decision types within this space. A full list of decisions is in Appendix H.

Decisions were instantiated within two contexts: one where all items are assumed to have a positive value (candies), and one with negative values (electric shocks). These meaningfully change the inferences of a observed decision; choosing {candy $A$} over {candy $B$, candy $C$} indicates a strong preference for $A$, but choosing {shock $A$} over {shock $B$, shock $C$} does not.

Through these rankings, Jern et al. (2017) found that humans ascribe almost perfectly rational decision-making models to the people they observe.

**Rational models.** Inverse decision-making models developed by psychologists to explain how people infer the preferences of others typically first specify a forward model and then infer preferences by applying Bayesian inference (Baker et al., 2009; Lucas et al., 2014; Jara-Ettinger et al., 2020; Ho and Griffiths, 2021). The forward model is normally a "noisy" version of a rational model, where options with greater utility are selected with higher probability. For example, given the utilities a decision-maker assigns to each item $\mathbf{u}$ and the items in each of the $n$ options $\mathbf{A} = \{a_1, \ldots, a_n\}$, a standard model based on Luce (1959) assumes the probability of choosing option $o_j$ in choice $c$ is

$$p(c = o_j | \mathbf{u}, \mathbf{A}) = \frac{\exp(U_j)}{\sum_{k=1}^{n} \exp(U_k)}, \tag{1}$$

where $U_j$ is the sum of utilities for all items in option $j$.[3]

To make rational inferences about the preferences (utilities $\mathbf{u}$) that motivated the observed choice, the posterior over utilities $p(\mathbf{u}|c, \mathbf{A})$ is inverted using Bayes' rule:

$$p(\mathbf{u}|c, \mathbf{A}) = \frac{p(c|\mathbf{u}, \mathbf{A})p(\mathbf{u})}{p(c|\mathbf{A})}. \tag{2}$$

Put simply, the posterior distribution $p(\mathbf{u}|c, \mathbf{A})$ is computed starting from a prior $p(\mathbf{u})$, scaled by the likelihood of making the choice $p(c|\mathbf{u}, \mathbf{A})$, and normalized by the marginal likelihood $p(c|\mathbf{A})$.

Given the utilities, when a rational agent reasons about which observed decision provides more evidence that a decision-maker prefers a certain item, Jern et al. (2017) suggest two prevailing theories that correlate higher with human behavior than others: absolute utility and relative utility. Absolute utility posits that the expected utility of an item $x$ over the posterior distribution directly corresponds to there being more evidence for the decision-maker preferring the item $x$:

$$\text{preference}(x) \propto \mathbb{E}[u_x | c, \mathbf{A}]. \tag{3}$$

Meanwhile, relative utility posits that the preference of an item corresponds to the probability that its utility is highest amongst all items:

$$\text{preference}(x) \propto p(\forall i, u_x > u_i | c, \mathbf{A}). \tag{4}$$

---

[3]We adopt the classic assumption that utilities of multiple items in an option are combined linearly. This may not be true in realistic scenarios, e.g., if someone has an ice cream they are less likely to want a lollipop.

After measuring the inferences that LLMs make about others' utilities based on observed decisions, we compare them against the predictions from the absolute utility and relative utility models. In addition, we also compare them against two components of the right-hand expression of Equation 2, the likelihood $p(c|\mathbf{u}, \mathbf{A})$ and the inverse of the marginal likelihood $1/p(c|\mathbf{A})$ (henceforth referred to as "marginal likelihood" for simplicity), which correspond to simpler — yet still rational — behavior. By themselves, these components have lower correlations with human behavior, but they serve as important building blocks for the both absolute and relative utility (Jern et al., 2017).

**Human data.** We compare LLMs' outputs against data collected from people performing the original task of ranking the 47 decisions, conducted by Jern et al. (2017). Jern et al. found that the rational models of absolute utility and relative utility both achieve Spearman correlations of 0.98 with human inferences, outperforming likelihood, marginal likelihood, and feature-based models.

**Language models.** We ran our LLM experiments on Llama-3-8B, Llama-3-70B, Claude 3 Opus, GPT-4-Turbo (0125-preview), and GPT-4o. Experiments were conducted in April and May of 2024. We set sample sizes of 43 for the positive context and 42 for the negative context, which are equal to the sample size of the original human experiment. This was obtained for all models aside from Claude 3 Opus, where we set an artificial sample size of 5 due to cost constraints. We used temperature = 1 across all models, and prompted using both zero-shot and chain-of-thought prompting. For each sample, we queried the LLM to make $\binom{47}{2}$ decisions, one for each pairwise comparison.

We constructed prompts based on the original scripts and text instructions given to participants in the human experiment, adapted to pairwise comparisons instead of ranking 47 decisions at once. We also removed physical details of the experiment (e.g., the decisions were printed on cards with colors to represent items, while we describe items with natural language). The prompt first introduces the context (either candy or electric shocks), describes the pair of observed decisions, and concludes with the request to select the choice that more strongly suggests that the decision-maker prefers the target item. We also included additional clarifications present in the original human experiment, as well as instructions for structuring the outputs (e.g., chain-of-thought) if applicable. To mitigate effects from LLMs' positional bias (Wang et al., 2023) and any potential context biases related to item descriptions, we shuffled the individual contexts we assigned to each item (e.g., black vs. blue candy). We also shuffled the order that decisions appear in the pair, shuffled the order of options within the decisions, and shuffled the order of items within each option. See Appendix G for examples.

After LLMs make the pairwise decisions, we parse the answers based on a handmade rule-based classifier. In the chain-of-thought case, if the classifier is unable to categorize the answer, we re-prompt the LLM asking it to classify its response. After we have results for all $\binom{47}{2}$ pairwise comparisons, we aggregate them into a ranking ordered by the number of pairwise wins (ties are considered 0.5). We then compare these rankings against those of humans and rational models.

## 4.1 RESULTS

To investigate the decision-making models behind how LLMs make inferences from observed decisions, we compare the Spearman correlation of LLMs' inferences against those made by humans and rational models. We organize the results into two main takeaways.

**Stronger LLMs become highly capable at rational modeling.** The inferences of LLMs have positive correlations with both absolute and relative utility, and that this correlation rises both as model capabilities improve and when models are allowed inference-time reasoning (see Table 4); for the positive CoT case, Llama-3-8B achieves 0.62 correlation with absolute and relative utility, while Llama-3-70B achieves correlations of {0.88, 0.89} and GPT-4o achieves correlations of {0.95, 0.94}.

Though LLMs may have been trained on data from the original experiment, our setup with pairwise decisions, extensive shuffling, and prompt adaptations ensure that LLMs' prior experiences with Jern et al. (2017) do not help it "cheat" and make more rational choices. Thus, we can attribute high correlations with rational models as evidence that LLMs implicitly assume rationality in this setting.

**LLM inferences are highly correlated with people.** We also find that LLMs have high correlations with the inferences made by people. GPT-4o with CoT achieves a 0.97 Spearman correlation with

Table 4: Spearman correlations between inverse decision rankings made by LLMs / humans and predictions of rational models. LLMs that most highly correlate with humans are in **bold**. Correlation coefficients with absolute value $\geq .3$ are statistically significant at $\alpha = .05$, and $\geq .47$ at $\alpha = .001$.

| context | prompt | compared with | Llama-3-8B | Llama-3-70B | Claude 3 Opus | GPT-4-Turbo | GPT-4o | humans |
|---|---|---|---|---|---|---|---|---|
| positive (candies) | CoT | humans | 0.66 | 0.92 | 0.92 | 0.95 | **0.97** | 1.00 |
| | | absolute utility | 0.62 | 0.88 | 0.89 | 0.93 | 0.95 | 0.98 |
| | | relative utility | 0.62 | 0.89 | 0.92 | 0.94 | 0.94 | 0.98 |
| | | likelihood | -0.57 | -0.51 | -0.43 | -0.42 | -0.45 | -0.51 |
| | | marginal likelihood | 0.67 | 0.70 | 0.66 | 0.66 | 0.70 | 0.76 |
| | zero-shot | humans | 0.28 | 0.63 | 0.56 | **0.65** | **0.65** | 1.00 |
| | | absolute utility | 0.20 | 0.57 | 0.52 | 0.59 | 0.59 | 0.98 |
| | | relative utility | 0.23 | 0.59 | 0.53 | 0.60 | 0.60 | 0.98 |
| | | likelihood | -0.62 | -0.68 | -0.56 | -0.52 | -0.74 | -0.51 |
| | | marginal likelihood | 0.57 | 0.72 | 0.62 | 0.58 | 0.77 | 0.76 |
| negative (shocks) | CoT | humans | 0.53 | 0.68 | 0.74 | 0.77 | **0.87** | 1.00 |
| | | absolute utility | 0.25 | 0.42 | 0.48 | 0.59 | 0.68 | 0.90 |
| | | relative utility | 0.40 | 0.57 | 0.59 | 0.63 | 0.74 | 0.93 |
| | | likelihood | -0.03 | 0.00 | 0.04 | 0.06 | -0.05 | -0.28 |
| | | marginal likelihood | 0.21 | 0.23 | 0.24 | 0.24 | 0.36 | 0.61 |
| | zero-shot | humans | 0.17 | 0.43 | 0.43 | **0.53** | 0.51 | 1.00 |
| | | absolute utility | -0.11 | 0.17 | 0.15 | 0.31 | 0.24 | 0.90 |
| | | relative utility | 0.03 | 0.28 | 0.29 | 0.36 | 0.31 | 0.93 |
| | | likelihood | -0.06 | -0.09 | -0.14 | 0.12 | -0.09 | -0.28 |
| | | marginal likelihood | 0.09 | 0.20 | 0.25 | 0.25 | 0.24 | 0.61 |

human behavior, indicating that it makes inferences about others that are extremely similar to those made by people. We also observe that like humans, LLMs are less consistent with the rational model in the more difficult negative context, and have negative correlations with the likelihood component of rational models. In the zero-shot positive case, LLMs seem to be much more correlated with marginal likelihood than the more complex rational models, indicating that it may be using this simpler decision-making model as a proxy when given no context to reason about its answer.

We also observe that LLMs' correlations with human inferences are consistently higher than their correlations with rational models. This is especially true in the negative context where humans are less consistent with the rational model (e.g., GPT-4o with CoT has a 0.87 correlation with humans, compared to a 0.74 correlation with the highest rational model). Thus, although LLMs typically assume rationality, when people's inferences diverge from those of rational models, LLMs' inferences are closer to humans. This could be explained by LLMs sharing some heuristic strategies with humans, a topic for future investigation. Scatterplots showing patterns of responses are in Appendix F.

## 5 DISCUSSION

We conducted an extensive evaluation of how LLMs assume people make decisions. In our forward modeling experiments, we found that LLMs struggle to predict or simulate human behavior in a classic risky choice setting, assuming that people make decisions more rationally than we actually do. We connect this to a previous finding in psychology — that people model others as more rational than they are — in order to explain why people think LLMs produce human-like behavior when making decisions. Then in our inverse modeling experiments, we find that LLMs also assume people act rationally when reasoning backwards from observed actions to internal utilities, aligning with how humans make inferences about others' choices. Thus, LLMs seem to adopt a consistent model of human decision-making across forward and inverse modeling — one that assumes people act more rationally than we actually do.

**Implications for aligning and training LLMs.** The psychology literature shows that there is a dichotomy between how people make decisions and how we expect others to make decisions. How should alignment be defined when these are different? Existing frameworks that focus on safe and useful deployments (e.g., Bommasani et al., 2021; Askell et al., 2021) may prioritize aligning with our expectations, but there are also many merits to having models behave like us (e.g., Park et al., 2022; Shaikh et al., 2024). We believe a reasonable answer to this is to separate alignment into two sub-cases: alignment with human expectations and alignment with human behavior, and to train

separate models that fulfill each objective. Models aligned with human expectation should shed human tendencies such as resource-rationality, i.e. sacrificing quality to reduce effort (Evans et al., 2015; Alanqary et al., 2021; Lieder and Griffiths, 2020), while models designed to simulate humans should retain them. By recognizing the difference between people's expectations and behavior, we provide support for developing more specific alignment objectives grounded in social science.

We hypothesize that certain training paradigms may be more suited towards aligning to human expectations, while others favor alignment with human behavior. For instance, high-quality written responses used in Supervised Fine-Tuning may teach the LLM to mimic the original human writers, aligning outputs towards human behavior. On the other hand, when humans provide preferences for RLHF in a chat setting, their judgement might reflect what the human rater expects from the LLM.

**Implications for simulating humans using LLMs.** There is a growing literature investigating whether we can use LLMs to simulate humans for various applications (Park et al., 2023), such as acting as mock participants in human studies (Argyle et al., 2023; Aher et al., 2022; Hämäläinen et al., 2023), collecting public opinion (Chu et al., 2023; Kim and Lee, 2023; Sun et al., 2024), and helping provide realistic reactions to assist people's communication (Liu et al., 2023; Shaikh et al., 2024; Lin et al., 2024; Shin and Kim, 2023). As our experiments show that LLMs make decisions more rationally than people do, and also predict people's decisions to be more rational than they are, current LLMs are fundamentally misaligned for the task of simulating human choices. Developing policy recommendations or designing experiments based on LLM-simulated choices may be misleading — similar to concerns about the use of an overly rational "homo economicus" originally raised by researchers in behavioral economics (Tversky and Kahneman, 1974; Kahneman and Tversky, 1979).

**Limitations.** Our experiments focused on a subset of all tasks for which we could evaluate models' rationality. This subset was carefully selected to be representative of the classic literature, but could be expanded upon in future research. In particular, our experiments used controlled, abstract domains that do not guarantee generalization to real-world contexts. While this is the same challenge faced by all human experiments, it is uniquely challenging for work done on LLMs due to their black-box nature and sensitivity to input prompts (Sclar et al., 2023). Furthermore, like all human data, the psychological datasets that we compare LLMs against were potentially subject to sampling bias and it is unknown whether they fully represent the true distribution of human choices or inferences.

Another limitation is that we use simple simulation paradigms for testing LLMs' capabilities. While we conduct an ablation using demographic prompts, other methods such as adding personal details or traits for more realistic variation have been proposed (Zhou et al., 2024b; Hu and Collier, 2024). The `choices13k` dataset does not contain such information about subjects, making a representative inclusion of these features impossible. However, future work could conduct a more comprehensive analysis of different simulation methods to see if they alter LLMs' implicit decision-making models.

A third limitation to our work is that we do not directly compare LLM predictions of human decisions with human predictions of the same decisions. The psychology literature has shown that there can be differences between when people perform the risky choice task and when they predict how others would perform the same task: Faro and Rottenstreich (2006) found that people's predictions of others' choices are closer to risk neutrality than choices of their own; when people are risk seeking, they predict that others will be risk seeking but less so; and where people are risk averse, they predict that others will be risk averse but also less so. Collecting a dataset of people's predictions about others' decisions could allow us to make a quantitative comparison of humans and LLMs in this setting.

**Future directions.** A peculiar observation from the inverse modeling setting is that LLMs' inferences were consistently more correlated with people than they were with rational models. This suggests that LLMs may potentially capture aspects of human behavior that are not present in existing theoretical models. While we have focused on using theories and paradigms from psychology to analyze LLMs, there may also be opportunities to use LLMs to refine existing theories about people. More generally, our results show how studies and theories originating in psychology and cognitive science can help quantify the behavior of LLMs. These fields offer many more opportunities to compare the behavior of LLMs with humans, helping us towards understanding (both) these complex yet extremely capable systems.

ACKNOWLEDGEMENTS

Experiments were supported by Azure credits from a Microsoft AFMR grant. This work and related results were made possible with the support of the NOMIS Foundation. Special thanks to Theodore R. Sumers, Jian-Qiao Zhu and Mengzhou Xia for providing invaluable feedback for this work.

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

# A    FORWARD MODELING PROMPTS

Prompts for forward modeling zero-shot are in Table 5, and prompts for forward CoT are in Table 6.

Table 5: Zero-shot prompting for forward modeling paradigms

---

**LLMs predict individual human participant's choice:**
A person is presented with two gambling machines, and makes a choice between the machines with the goal of maximizing the amount of dollars received.

The person will get one reward from the machine they choose. A fixed proportion of 10% of this value will be paid to the participant as a performance bonus.
If the reward is negative, their bonus is set to $0.

Machine A: {}
Machine B: {}

Which machine does the person choose?
Do not provide any explanation, only answer with A or B:

---

**LLMs predict the human choice distribution:**
{} people are presented with two gambling machines, and each person makes a choice between the machines with the goal of maximizing the amount of dollars received.
Each person will get one reward from the machine they choose. A fixed proportion of 10% of this value will be paid to the participant as a performance bonus.
If the reward is negative, their bonus is set to $0.

Machine A: {}
Machine B: {}

How many people choose Machine A?
How many people choose Machine B?

Please only provide the percentage of people who choose Machine A and Machine B in the json format.

---

**LLMs act as human participant:**
There are two gambling machines, A and B. You need to make a choice between the machines with the goal of maximizing the amount of dollars received.
You will get one reward from the machine that you choose. A fixed proportion of 10% of this value will be paid to you as a performance bonus.
If the reward is negative, your bonus is set to $0.

Machine A: {}
Machine B: {}

Which machine do you choose?
Do not provide any explanation, only answer with A or B:

---

# B    FORWARD MODELING RESULTS WITH DIFFERENT PROMPTS

To investigate how LLMs estimate human behavior for the overall sample sizes, we asked LLMs to predict the probability distribution of human choices between gamble machine A and gamble machine B. We observed that estimate the probablity of the overall decisions mitigates the strong correlation with the maximum expected value of each machine while not improving the correlation with actual human behaviors. Particularly for zero-shot prompting, both closed-source and open-source models in Table 7 show a drop in correlation values between 14% and 25% compared to the zero-shot prompting results in Table 1.

We also observed that even when LLMs are asked to act as human participants in making decisions, their outcomes remain consistently more rational than actual human behavior under CoT prompting. Table 10 shows the results of LLMs' decisions as individual human participants compared to the maximum expected value. Both GPT-4-Turbo and GPT-4o exhibit a high correlation with the maximum expected value, each with a correlation coefficient of 0.91 and 0.92 and the MSE of 0.045 and 0.037, while still maintaining a moderate correlation with actual human behavior, as shown in Table 9.

Table 6: CoT prompting for forward modeling paradigms

---

**LLMs predict individual human participant's choice:**
A person is presented with two gambling machines, and makes a choice between the machines with the goal of maximizing the amount of dollars received.

The person will get one reward from the machine they choose. A fixed proportion of 10% of this value will be paid to the participant as a performance bonus.
If the reward is negative, their bonus is set to $0.

Machine A: {}
Machine B: {}

Which machine does the person choose?
Let's think step by step before answering with A or B:

---

**LLMs predict the human choice distribution:**
{} people are presented with two gambling machines, and each person makes a choice between the machines with the goal of maximizing the amount of dollars received.
Each person will get one reward from the machine they choose. A fixed proportion of 10% of this value will be paid to the participant as a performance bonus.
If the reward is negative, their bonus is set to $0.

Machine A: {}
Machine B: {}

How many people choose Machine A?
How many people choose Machine B?

Let's think step by step before providing the final output.
Please provide the percentage of people who choose Machine A and Machine B in the json format.

---

**LLMs act as human participant:**
There are two gambling machines, A and B. You need to make a choice between the machines with the goal of maximizing the amount of dollars received.
You will get one reward from the machine that you choose. A fixed proportion of 10% of this value will be paid to you as a performance bonus.
If the reward is negative, your bonus is set to $0.

Machine A: {}
Machine B: {}

Which machine do you choose?
Let's think step by step before answering with A or B:

---

Table 7: The correlation between LLMs predicting the human choice distribution based on the aggregate sample size of participants and the actual human choice.

|  |  | Llama3-8B | Llama3-70B | GPT-4-Turbo | GPT-4o | Humans |
|---|---|---|---|---|---|---|
| Zero-shot | Spearman correlation | 0.1045 | 0.2827 | 0.4812 | 0.6156 | / |
|  | Pearson correlation | 0.1032 | 0.2904 | 0.4830 | 0.6112 | / |
|  | MSE | 0.2811 | 0.2668 | 0.1951 | 0.1633 | / |
| CoT | Spearman correlation | 0.1799 | 0.1046 | 0.6208 | 0.5825 | / |
|  | Pearson correlation | 0.1783 | 0.0992 | 0.6308 | 0.6012 | / |
|  | MSE | 0.2615 | 0.2954 | 0.1202 | 0.1282 | / |

## C  MODEL CORRELATIONS FOR FORWARD MODELING

We provide the full correlation results for the three forward modeling paradigms for Llama3-8B, Llama3-70B, GPT-4-Turbo (0125-preview), and GPT-4o in Figure 2. Compared to the correlations in zero-shot prompting, CoT prompting shows a higher degree of correlation across all four LLMs.

## D  COMPARING LLMS TO A WIDER RANGE OF BEHAVIORAL MODELS

Our results clearly show that chain-of-thought results in LLM responses that align closely with expected value. To try to understand whether there was a systematic pattern in the responses of the

Table 8: The correlation between LLMs predicting the human choice distribution based on the aggregate sample size of participants and the maximum expected value.

|  |  | Llama3-8B | Llama3-70B | GPT-4-Turbo | GPT-4o | Humans |
|---|---|---|---|---|---|---|
| Zero-shot | Spearman correlation | 0.0426 | 0.1688 | 0.3380 | 0.1741 | 0.4835 |
| | Pearson correlation | 0.0426 | 0.1688 | 0.3380 | 0.1741 | 0.4835 |
| | MSE | 0.4752 | 0.4361 | 0.3465 | 0.4527 | 0.2580 |
| CoT | Spearman correlation | 0.2025 | 0.8406 | 0.8458 | 0.8518 | 0.4835 |
| | Pearson correlation | 0.2025 | 0.8406 | 0.8458 | 0.8518 | 0.4835 |
| | MSE | 0.3978 | 0.0807 | 0.0726 | 0.0702 | 0.2580 |

Table 9: The correlation between LLMs acting as a human participant to make choice and the actual human choice.

|  |  | Llama3-8B | Llama3-70B | GPT-4-Turbo | GPT-4o | Humans |
|---|---|---|---|---|---|---|
| Zero-shot | Spearman correlation | 0.4047 | 0.4528 | 0.5841 | 0.4617 | / |
| | Pearson correlation | 0.4068 | 0.4559 | 0.5667 | 0.4565 | / |
| | MSE | 0.0920 | 0.1372 | 0.1414 | 0.2031 | / |
| CoT | Spearman correlation | 0.4597 | 0.6223 | 0.6153 | 0.6074 | / |
| | Pearson correlation | 0.4600 | 0.6165 | 0.6115 | 0.6030 | / |
| | MSE | 0.1972 | 0.1621 | 0.1640 | 0.1659 | / |

Table 10: The correlation between LLMs acting as a human participant to make choice and the maximum expected value.

|  |  | Llama3-8B | Llama3-70B | GPT-4-Turbo | GPT-4o | Humans |
|---|---|---|---|---|---|---|
| Zero-shot | Spearman correlation | 0.1774 | 0.3130 | 0.4190 | 0.2944 | 0.4835 |
| | Pearson correlation | 0.1730 | 0.3069 | 0.4053 | 0.2944 | 0.4835 |
| | MSE | 0.2888 | 0.2980 | 0.2729 | 0.3536 | 0.2580 |
| CoT | Spearman correlation | 0.5155 | 0.8353 | 0.9100 | 0.9255 | 0.4835 |
| | Pearson correlation | 0.5155 | 0.8353 | 0.9100 | 0.9255 | 0.4835 |
| | MSE | 0.2492 | 0.0836 | 0.0450 | 0.0372 | 0.2580 |

zero-shot models, we fit 18 choice models from the behavioral sciences to the output of GPT-4 using the zero-shot individual choice prompt. The set of models was based on those used in Peterson et al. (2021), where they are described in more detail together with references to the original paper.

In this context, each model represents a hypothesis about the LLM's beliefs about human behavior. These included Heuristic models (He et al., 2022), wherein people are thought to emply mental shortcuts to make decisions, Counterfactual models (He et al., 2022), which appeal to constructs like regret and disappointment, and Subjective Expected Utility models (He et al., 2022), which assume that quantities involved (money and probability) are perceived or otherwise treated subjectively. This latter, third category contains many of the most influential models—Expected Utility Theory (EU) and Prospect Theory (PT)—as well as the model proposed by Peterson et al. (2021) which is called Mixture of Theories (MOT).

Table 11 shows the results. Models in the top two sections of the table (Heuristic and Counterfactual) provided strictly inferior fits compared to Subjective Expected Utility models in the third section. Among those in the third section, Expected Value provided the worst fit. Expected Utility was notably better, suggesting that GPT-4 correctly assumes that people do not treat the value of money objectively / linearly. Prospect Theory improved this score slightly through the incorporation of a subjective probability weighting function, but that fitted function was largely linear, suggesting that GPT-4 incorrectly assumes that people do not treat probabilities subjectively. Lastly, MOT provided the best fit to the inferences of GPT-4. In previous work, MOT also provides the best fit to human data, but the fitted parameters are different (Peterson et al., 2021). When fitted directly to `choices13k`, MOT learns a mixture of two utility functions (e.g., like the one in Expected Utility) and two probability

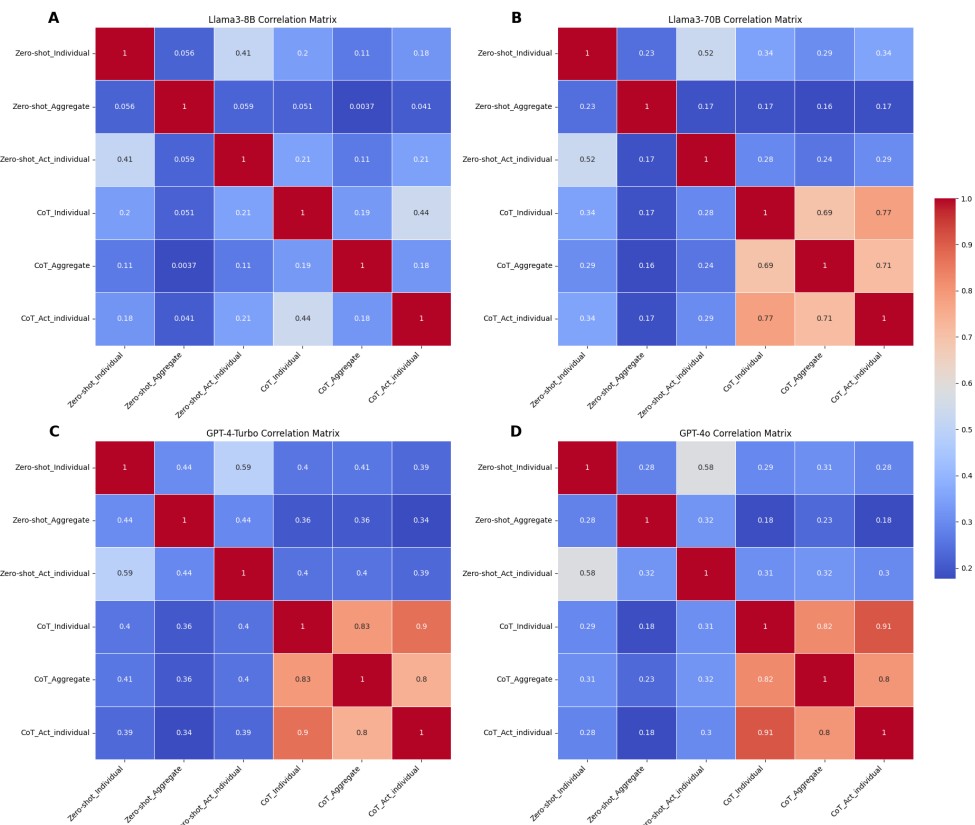

Figure 2: The correlations between LLMs [Llama3-8B, Llama3-70B, GPT-4-Turbo (0125-preview), GPT-4o]

weighting functions (e.g., like the one in Prospect Theory). Notably, one of the probability weighting functions is usually linear, and the other S-shaped. In the present case, one of the weighting functions was linear, but the other approximated a flat line. This suggests that GPT-4 expected people to be approximately rational most of the time, but completely ignores probabilities (i.e., weights them equally) in a minority of cases.

# E   ABLATION STUDY FOR FORWARD MODELING

## E.1   DIFFERENT TEMPERATURES

Originally, we chose temperature = 1 for some tasks and temperature = 0.7 for others. While 1 is the default temperature used by most models, 0.7 is another common option, used by works such as AlpacaEval (Li et al., 2023) on instruction following tasks.

To further validate the robustness of our findings, we test the forward modeling experiment using GPT-4o with temperatures 0.0, 0.5, 1.0, 1.5, and 2.0 for both zero-shot and CoT conditions, and compute spearman correlation with both human and max expected value, shown in Tables 12 and 13. We find that varying temperature does not substantially alter our findings other than when conducting CoT with high temperatures, where models become unable to circle back to an answer due to noisy outputs.

Table 11: MSE between GPT-4-individual-zero-shot outputs and the fitted predictions of behavioral models.

| Behavioral Model | MSE |
|---|---|
| Better Than Average | 0.20473 |
| Equiprobable | 0.20212 |
| Low Payoff Elimination | 0.18248 |
| Low Expected Payoff Elimination | 0.18383 |
| Probable | 0.20559 |
| Minimax | 0.20751 |
| Maximax | 0.20401 |
| Priority Heuristic | 0.18994 |
| Disappointment Theory with EV | 0.16125 |
| Disappointment Theory with EU | 0.12273 |
| Disappointment Theory Without Rescaling | 0.16134 |
| RegretTheory with EV | 0.15918 |
| RegretTheory with EU | 0.12278 |
| Expected Value | 0.16134 |
| Expected Utility | 0.11435 |
| Prospect Theory | 0.11427 |
| Transfer of Attention Exchange | 0.12028 |
| Mixture of Theories | **0.09835** |

Table 12: Comparison of Zero-shot vs. Max Expected Value and CoT vs. Max Expected Value across temperature settings for GPT-4o.

| Zero-shot vs. Max Expected Value | | CoT vs. Max Expected Value | |
|---|---|---|---|
| Temp | Spearman | Temp | Spearman |
| 0.0 | 0.313 | 0.0 | 0.540 |
| 0.5 | 0.272 | 0.5 | 0.515 |
| 1.0 | 0.284 | 1.0 | 0.539 |
| 1.5 | 0.302 | 1.5 | 0.351 |
| 2.0 | 0.308 | 2.0 | *model unable* |

## E.2 DEMOGRAPHIC GROUPS FOR DECISION MAKERS

To alleviate concerns about the simplicity of the experimental setup and further validate our findings, we conducted a follow-up experiment replicating our original analysis on various demographic profiles with varying age and gender (Tables 14 and 15). We find that our analyses stay consistent across all these factors, suggesting that our findings are at least somewhat robust to variations in context and prompt, as well as differing profiles of people.

## E.3 LESS RATIONAL DECISION MAKERS

For additional prompt variation, we also prompt the LLM to make the same predictions for a monkey rather than a human (Tables 14 and 15, bottom row). When performing both zero-shot and CoT, models accurately reflect that monkeys are less rational and the resulting predictions are less correlated with max expected value. However, for the CoT condition, the correlations are also increased with actual human behavior. This supports our conclusions with an example where the model predicts less rational decisions, but these are actually more in-line with how humans actually behave.

## F FULL RESULTS FOR INVERSE MODELING

In this section, we provide the correlation plots between humans/rational models and LLMs. For brevity, we consider only the positive CoT context. The ordering of the plots begins with GPT-4o, followed by GPT-4 Turbo, Claude-3 Opus, Llama-3-70B, and concludes with Llama-3-8B.

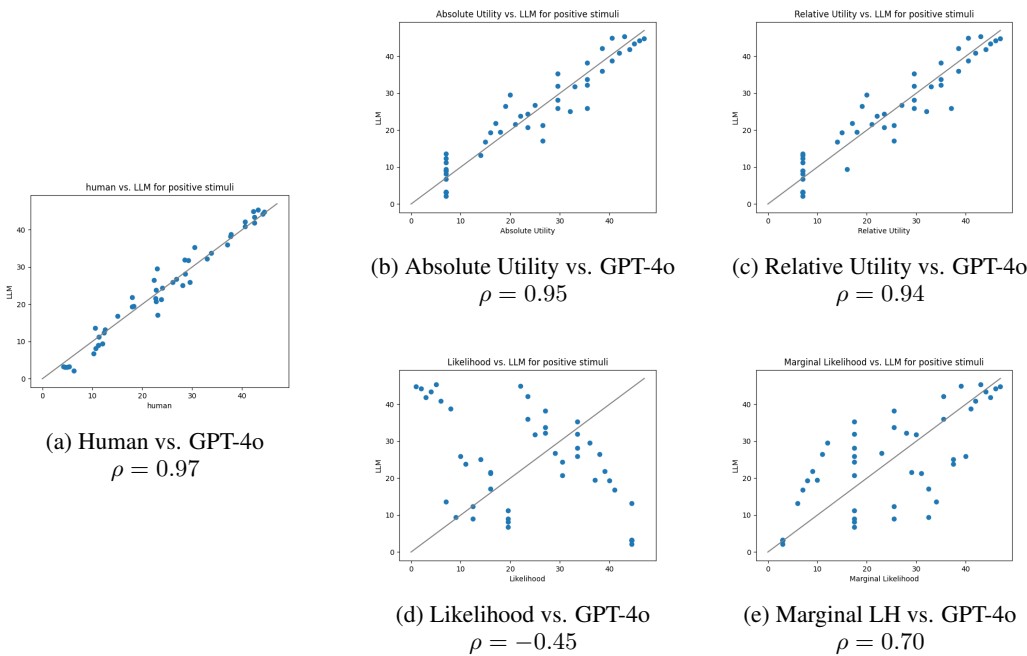

Figure 3: Comparing GPT-4o CoT rankings (y-coordinates) to humans and four theoretical decision-making models (x-coordinates) in positive setting.

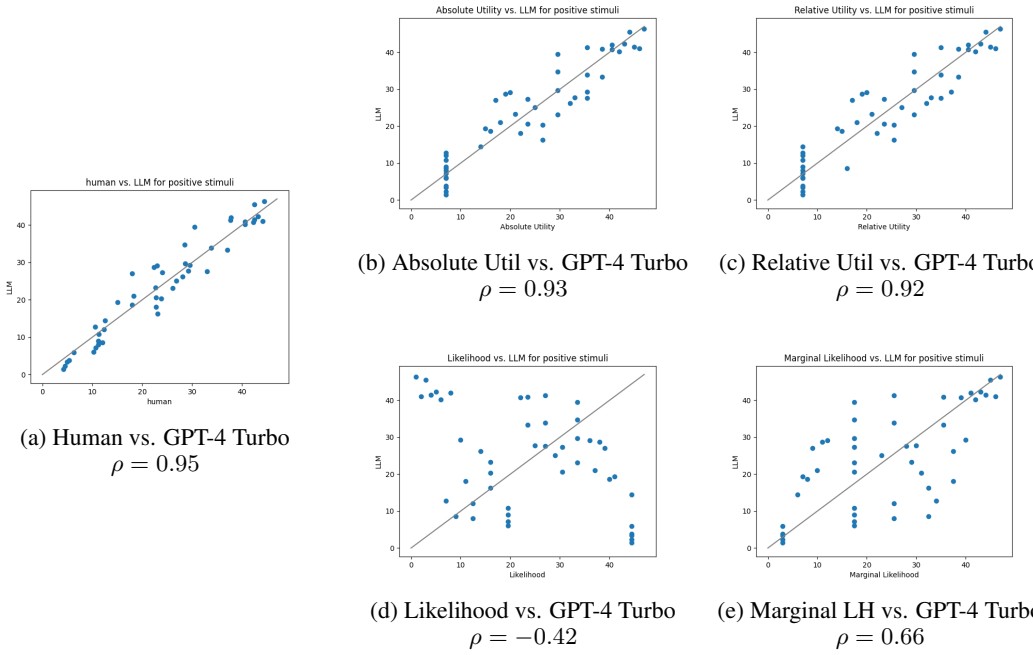

Figure 4: Comparing GPT-4 Turbo (0125-preview) CoT rankings (y-coordinates) to humans and four theoretical decision-making models (x-coordinates) in positive setting.

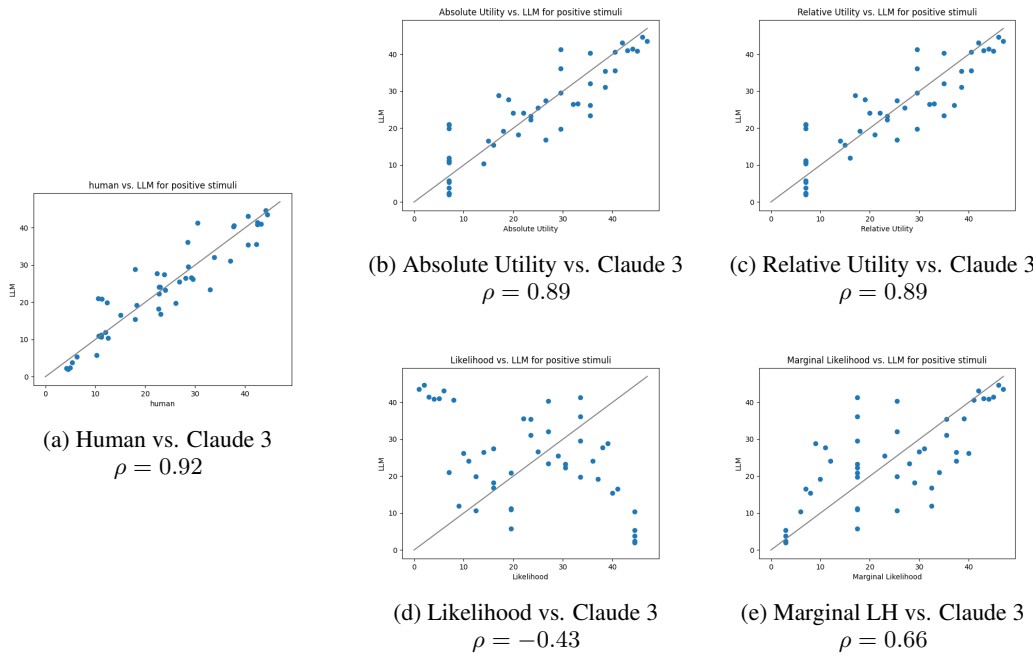

Figure 5: Comparing Claude 3 Opus CoT rankings (y-coordinates) to humans and four theoretical decision-making models (x-coordinates) in positive setting.

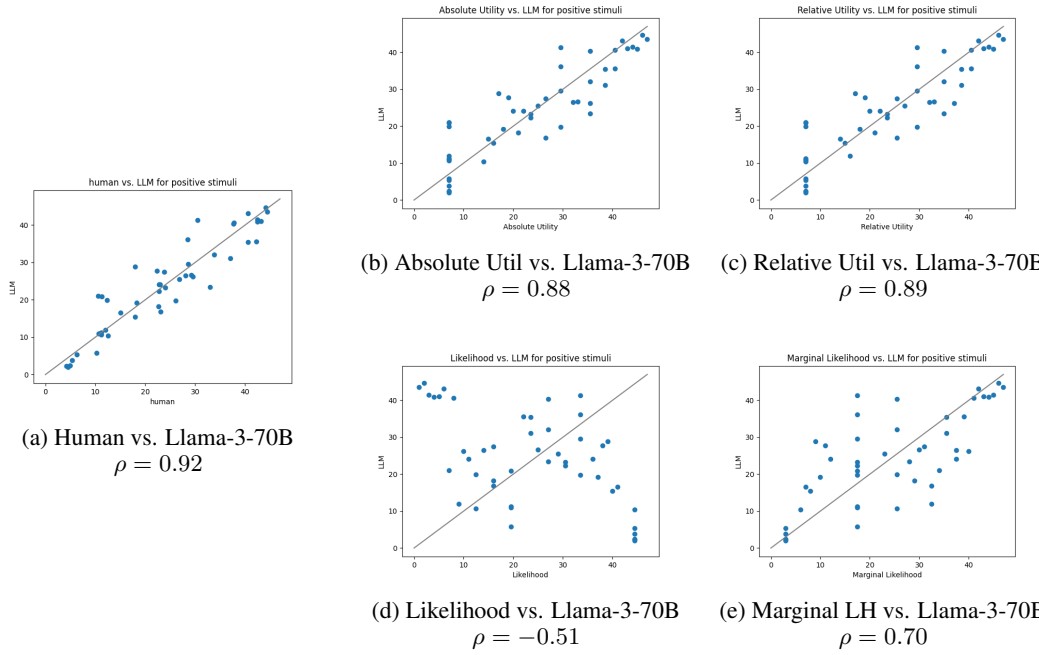

Figure 6: Comparing Llama-3-70B CoT rankings (y-coordinates) to humans and four theoretical decision-making models (x-coordinates) in negative setting.

Table 13: Comparison of Zero-shot vs. Human and CoT vs. Human across temperature settings for GPT-4o.

| Zero-shot vs. Human | | CoT vs. Human | |
|---|---|---|---|
| Temp | Spearman | Temp | Spearman |
| 0.0 | 0.533 | 0.0 | 0.540 |
| 0.5 | 0.381 | 0.5 | 0.515 |
| 1.0 | 0.341 | 1.0 | 0.518 |
| 1.5 | 0.489 | 1.5 | 0.528 |
| 2.0 | *model unable* | 2.0 | *model unable* |

Table 14: Comparison of Zero-shot vs. Max Expected Value and CoT vs. Max Expected Value.

| Zero-shot vs. Max Expected Value | | | CoT vs. Max Expected Value | | |
|---|---|---|---|---|---|
| Demographic | Spearman | p-value | Demographic | Spearman | p-value |
| Woman 18-35 | 0.289 | 9.44E-21 | Woman 18-35 | 0.970 | <1E-150 |
| Woman 45-60 | 0.313 | 4.23E-24 | Woman 45-60 | 0.972 | <1E-150 |
| Woman 65-85 | 0.285 | 9.91E-21 | Woman 65-85 | 0.969 | <1E-150 |
| Man 18-35 | 0.335 | 2.50E-27 | Man 18-35 | 0.971 | <1E-150 |
| Man 45-60 | 0.308 | 3.37E-23 | Man 45-60 | 0.965 | <1E-150 |
| Man 65-85 | 0.348 | 3.98E-29 | Man 65-85 | 0.964 | <1E-150 |
| Monkey | 0.208 | 3.49E-13 | Monkey | 0.867 | 7.37E-123 |

## G    INVERSE MODELING PROMPTS

Example inverse modeling prompts for zero-shot and CoT are shown in Tables 16 and 17. First, the context of the experiment is introduced, then the choices are listed, and lastly the LLM is asked to reply with which comparison more strongly suggests that the decision-maker prefers a certain target item.

## H    47 DECISIONS USED IN INVERSE DECISION-MAKING EXPERIMENT

We provide a list of the 47 decisions used in the inverse decision-making experiment of Jern et al. (2017) in Table 18. Columns represent options, and letters represent items with the options. Based on the context, letters were replaced with colored candies or numbered electric shocks. Participants ranked these decisions by their strength in suggesting that the decision-maker preferred item $x$ over the other items.

Table 15: Comparison of Zero-shot vs. Human and CoT vs. Human.

| Zero-shot vs. Human | | | CoT vs. Human | | |
|---|---|---|---|---|---|
| Demographic | Spearman | p-value | Demographic | Spearman | p-value |
| Woman 18-35 | 0.529 | 2.75E-73 | Woman 18-35 | 0.549 | 9.23E-80 |
| Woman 45-60 | 0.519 | 6.07E-70 | Woman 45-60 | 0.546 | 3.54E-79 |
| Woman 65-85 | 0.520 | 2.75E-73 | Woman 65-85 | 0.550 | 3.70E-80 |
| Man 18-35 | 0.499 | 4.18E-64 | Man 18-35 | 0.528 | 2.10E-78 |
| Man 45-60 | 0.530 | 1.43E-73 | Man 45-60 | 0.539 | 1.34E-76 |
| Man 65-85 | 0.508 | 8.54E-67 | Man 65-85 | 0.531 | 3.57E-74 |
| Monkey | 0.482 | 2.18E-59 | Monkey | 0.630 | 8.47E-112 |

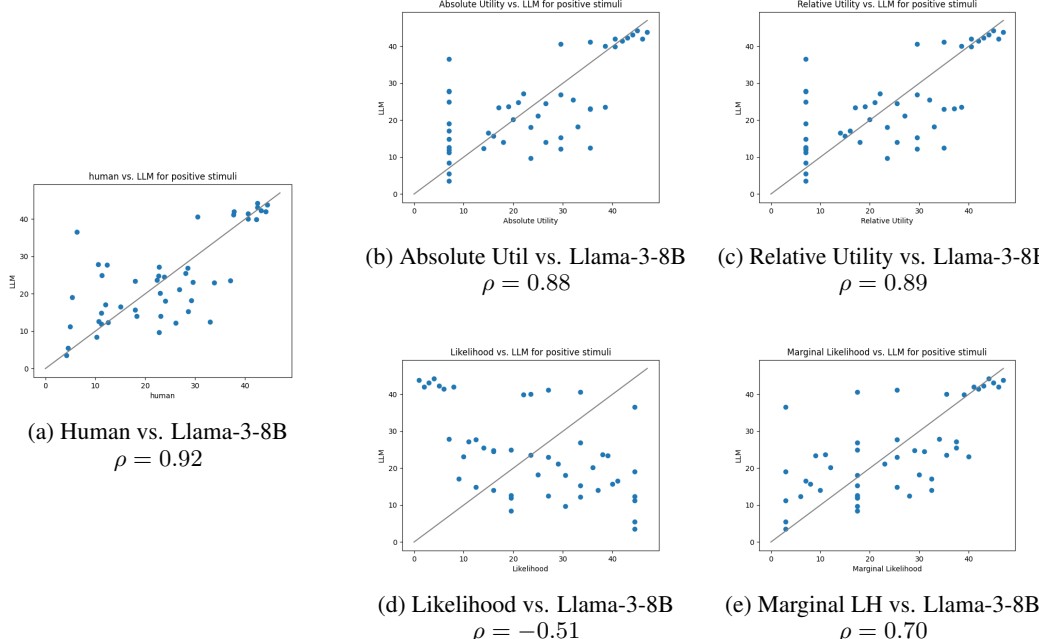

(a) Human vs. Llama-3-8B
$\rho = 0.92$

(b) Absolute Util vs. Llama-3-8B
$\rho = 0.88$

(c) Relative Utility vs. Llama-3-8B
$\rho = 0.89$

(d) Likelihood vs. Llama-3-8B
$\rho = -0.51$

(e) Marginal LH vs. Llama-3-8B
$\rho = 0.70$

Figure 7: Comparing Llama-3-8B CoT rankings (y-coordinates) to humans and four theoretical decision-making models (x-coordinates) in negative setting.

Table 16: Example prompt for inverse modeling, zero shot.

**Inverse Modeling, zero-shot:**
The following are two choices that people have made between different bags of candy. Each candy is a different color.
Choice 1 was made between the following bags:
Bag 1: red, brown, yellow, blue.
Bag 2: black.

The person making the choice chose Bag 2.

Choice 2 was made between the following bags:
Bag 1: yellow, black, red, brown.

The person making the choice chose Bag 1.

People were required to choose among the bags available, and were not allowed to reject all the bags.
For example, when there is only one bag, the person has no choice but to choose it.
Which choice (1 or 2) more strongly suggests that the person making the choice likes black candies?
Please respond with either "Choice 1" or "Choice 2". Do not include anything else in your answer.

Table 17: Example prompt for inverse modeling, chain-of-thought.

**Inverse Modeling, CoT:**
The following are two choices that people have made between different bags of candy. Each candy is a different color.
Choice 1 was made between the following bags:
Bag 1: red, brown, yellow, blue.
Bag 2: black.

The person making the choice chose Bag 2.

Choice 2 was made between the following bags:
Bag 1: yellow, black, red, brown.

The person making the choice chose Bag 1.

People were required to choose among the bags available, and were not allowed to reject all the bags.
For example, when there is only one bag, the person has no choice but to choose it.
Which choice (1 or 2) more strongly suggests that the person making the choice likes black candies?
Let's think step by step.

Table 18: List of 47 observed decisions from the inverse decision-making experiment of Jern et al. (2017). Decisions contained between 1-5 options, and each option corresponds to a column. The option in the leftmost column was chosen in all decisions. No options were empty; blank entries indicate that the decision had less than the maximum number of options. Each item is represented using a letter, with $x$ being the target item that inferences are made upon.

| option 1 | option 2 | option 3 | option 4 | option 5 |
|---|---|---|---|---|
| $d, c, b, a, x$ | | | | |
| $c, b, a, x$ | | | | |
| $b, a, x$ | | | | |
| $a, x$ | | | | |
| $x$ | | | | |
| $c, b, a, x$ | $d, b, a, x$ | | | |
| $a, x$ | $b, x$ | $c, x$ | $d, x$ | |
| $b, a, x$ | $c, a, x$ | | | |
| $b, a, x$ | $b, c, x$ | $b, d, x$ | | |
| $b, a, x$ | $d, c, x$ | | | |
| $a, x$ | $b, x$ | | | |
| $b, a, x$ | $c, a, x$ | $b, d, x$ | | |
| $a, x$ | $b, x$ | $c, x$ | | |
| $c, b, a, x$ | $d$ | | | |
| $b, a, x$ | $c$ | | | |
| $a, x$ | $b$ | | | |
| $b, a, x$ | $c$ | $d$ | | |
| $b, a, x$ | $d, c$ | | | |
| $a, x$ | $b$ | $c$ | | |
| $a, x$ | $b, x$ | $d, c$ | | |
| $b, a, x$ | $b, d, c$ | | | |
| $a, x$ | $b, x$ | $c, x$ | $a, d$ | |
| $a, x$ | $b$ | $c$ | $d$ | |
| $b, a, x$ | $b, c, x$ | $b, a, d$ | | |
| $a, x$ | $b, x$ | $a, c$ | | |
| $a, x$ | $c, b$ | | | |
| $c, b, a, x$ | $c, b, a, d$ | | | |
| $a, x$ | $b$ | $d, c$ | | |
| $a, x$ | $b, x$ | $a, c$ | $a, d$ | |
| $a, x$ | $a, b$ | | | |
| $b, a, x$ | $b, a, c$ | | | |
| $a, x$ | $a, b$ | $d, c$ | | |
| $a, x$ | $d, c, b$ | | | |
| $x$ | $a$ | | | |
| $b, a, x$ | $b, a, c$ | $b, a, d$ | | |
| $a, x$ | $a, b$ | $a, c$ | | |
| $a, x$ | $a, b$ | $a, c$ | $a, d$ | |
| $x$ | $a$ | $b$ | | |
| $x$ | $a$ | $b$ | $c$ | |
| $x$ | $a$ | $c, b$ | | |
| $x$ | $a$ | $b$ | $c$ | $d$ |
| $x$ | $c, b, a$ | | | |
| $x$ | $b, a$ | $d, c$ | | |
| $x$ | $a$ | $b$ | $d, c$ | |
| $x$ | $a$ | $d, c, b$ | | |
| $x$ | $d, c, b, a$ | | | |

