# OpenReview forum: "Large Language Models Assume People are More Rational than We Really are"
_ICLR.cc/2025/Conference — ICLR 2025 Poster_

### Official Review · Reviewer_dcdy · 2024-10-28

**Soundness:** 3
**Presentation:** 4
**Contribution:** 3
**Rating:** 8
**Confidence:** 4

**Summary:**

This paper compares the implicit decision-making process of LLMs with the human decision making process and finds that LLMs assume that people are more rational than they are, which is in line with what humans expect from other humans.

The paper uses two test suites: (1) Forward Modeling, which is a task in which the LLM needs to predict how likely people would select gamble A over gamble B, (2) Inverse Modeling, in which the LLM needs to predict people’s preferences from a list of choices. The authors compare the outputs of several LLMs with reported human answers to come to their conclusions.

The authors argue that the results have important implications for alignment, namely that alignment should be divided into two cases: alignment with human expectations, and alignment with human behavior.

**Strengths:**

* **Topic is important.** It is important to get insight in the decision process of LLMs, which has important consequences for how trustworthy we deem their outputs. This paper approaches this question from a CogSci angle.
* **Paper invites follow-up work.** The paper sets the stage for follow-up work on alignment. What it means to align well with humans, depends on whether one wants to align to human expectations, or human behavior, according to the authors.
* **Experiments are mostly solid.** The authors conduct experiments over a wide variety of popular LLMs, and compare the results. This gives insights in how the results generalize over different LLMs. Moreover, the experimental setup is strongly grounded in well-known experiments in Cognitive Science.

**Weaknesses:**

* **Unclear what the effect of the temperature setting is.** The authors briefly mention their choices for different temperature settings, but not in a lot of depth. Currently it is unclear how robust the findings are against different temperature settings (which would affect the outcomes). See questions below.
* **Unclear what the effect of different prompting strategies is.** The authors report the zero-shot prompt used, and share some of their considerations for designing the prompt. However, as LLMs can be sensitive to the actual prompt used, it would be good to understand how robust the findings are when different prompts are used, beyond the zero-shot vs. CoT. See questions below.

**Questions:**

Based on the weakness above:

* What does the CoT prompt look like? Only the zero-shot prompt is reported in the appendix.
* What is the effect of changing the temperature? Especially, how do the most popular temperature settings (when known) compare to temperature settings that are less common to use?
* What is the effect of changing the prompt? Did the authors try many different prompts to arrive at their results, or do they think the results are stable if one changes the prompt?

---

> ### Author Response · Authors · 2024-11-25
> **Response to Reviewer dcdy**
>
> Thank you so much for your review! We are humbled that you find our paper's topic important and impactful. We have carefully addressed each of your weakness / questions below, please take a look.
>
> ## Weakness 1 + Question 2:
> “Currently it is unclear how robust the findings are against different temperature settings (which would affect the outcomes).”
>
> We add an additional experiment varying the temperatures for forward modeling experiment in both zero-shot and CoT prompting cases. Please see item 2 in the general reviewer response!
>
> ## Weakness 2 + Question 3:
> “as LLMs can be sensitive to the actual prompt used, it would be good to understand how robust the findings are when different prompts are used, beyond the zero-shot vs. CoT.”
>
> We added an additional experiment utilizing demographic prompts across {age, gender} as well as a sanity check prompt which asks the LLM to predict the response for monkeys instead of humans.
>
> In these experiments, we found that results are consistent across the various demographic prompts, indicating that our findings are robust. LLM predictions are also less rational for monkeys, indicating that the model understands the prediction task and that it makes predictions based on the subject. See item 3 in the general reviewer response for details!
>
> ## Question 1:
> “What does the CoT prompt look like?”
>
> Thank you for pointing this out! We have added these into the appendix at Tables 6 and 13! For the record, chain-of-thought prompts replaced
> * “Please respond with either \"Choice 1\" or \"Choice 2\". Do not include anything else in your answer.”
>
> and
> * “Do not provide any explanation, only answer with A or B:”
>
> with
> * “Let's think step by step.”
>
> and
> * “Let’s think step by step before answering with A or B:”
>
> Please let us know if you have any other questions!

---

> > ### Comment · Reviewer_dcdy · 2024-11-25
> > **Thank you!**
> >
> > Thank you for answering my questions and providing these additional experiments.
> > Would be great to see those in the CR as well.
> >
> > Thanks for your engagement!

---

### Official Review · Reviewer_HdFJ · 2024-11-04

**Soundness:** 3
**Presentation:** 3
**Contribution:** 2
**Rating:** 6
**Confidence:** 4

**Summary:**

This paper investigates how LLMs perceive and predict human decision-making. The authors claim that LLMs (e.g., GPT-4 and Llama-3) tend to assume more rationality than people exhibit in reality. They adapt existing design of psychology experiments regarding human decisions, and compared human behavior with LLM predictions in two decision-making tasks: (1) predicting binary decision for gambling and (2) pairwise ranking the decisions regarding how much it tells about the human preference of an object. The experiments show that LLMs are more correlated with rational decisions than actual human behavior. The authors also claim this discrepancy between LLM rational assumptions and actual human behavior may have implications for alignment in AI systems, as well as for designing models that either simulate human behavior or align with human expectations.

**Strengths:**

- The authors explore a very interesting direction and implication in the current LLM alignment research. The current LLM alignment may actually limit the models’ ability to understand genuine human behavior.
- The authors link their work nicely to existing well-grounded cognitive science studies, which will be helpful to other readers who are not familiar with this area.

**Weaknesses:**

- While I also believe LLMs struggle to model irrationality, I’m not entirely convinced that the current experiment design fully supports this claim.
    - In the forward modeling section, the binary choice questions have an objectively optimal answer. Why should we conclude that the models are “assuming humans are more rational” simply because their answers are closer to this optimal? A simpler interpretation could be that the models are simply solving problems optimally, regardless of the prompt’s content.
    - The fact that the forward modeling part relies on binary choice questions amplified my concern. For instance, consider a set of binary math questions that are challenging for humans, where we ask the model to predict which answer a person might choose. If the model consistently picks the correct answer more often than humans, would it be valid to conclude that the model assumes humans are more rational than they actually are? Or is the model simply aiming to answer correctly?
- The forward modeling results in Table 1 & 2 indicate that models are more aligned with human choices than with maximum expected values in the zero-shot setting, but become more aligned with maximum expected values only when using CoT. Why, then, does this lead to the conclusion that “LLMs assume people are more rational than we really are”?

**Questions:**

- How do you measure the correlation for task 2 and 3 in the forward modeling task?
- Could there be circular inconsistencies in the pairwise ranking for the inverse modeling task? For instance, if the model ranks decision A > B, B > C, but then C > A, how is this resolved?
- Why not use “average human participant” for the prompts in Table 5 in the Appendix?
- Why did you decide to use “you” and refer this to human simulation? “What would an (average) human participant do?” sounds better?
- In Table 11 in the Appendix, where is Bag 2 in Choice 2? Is it referring to the Bag 2 from above?

---

> ### Author Response · Authors · 2024-11-25
> **Response to Reviewer HdFJ**
>
> Thank you so much for your review! We appreciate that you found our paper's findings interesting, especially in the scope of current alignment research! We have carefully responded to each of your weakness / questions below.
>
> ## Weakness 1:
> “In the forward modeling section, the binary choice questions have an objectively optimal answer. Why should we conclude that the models are “assuming humans are more rational” simply because their answers are closer to this optimal? A simpler interpretation could be that the models are simply solving problems optimally, regardless of the prompt’s content.”
>
> Thank you for the question! We agree that by just looking at the original results, it could also be possible that the model is solving the problem optimally regardless of the prompt. In this case, the model would be simply bad at understanding the problem context and instruction following.
>
> To test this alternative explanation, we conduct an experiment where we prompt the model to estimate the decision of monkeys instead of humans. This is shown in item 3 of the general response – “Additional experiment: Participant Groups and Prompting Variations”. With CoT, the correlation with max expected value of monkeys is only 0.65, which is much lower than the 0.95+ that we get for all other prompts. This suggests that the model isn’t simply computing the optimal solution.
>
> ## Weakness 2:
> “models are more aligned with human choices than with maximum expected values in the zero-shot setting”
>
> This is also a very good point! We realized we didn’t provide a very compelling argument for why zero-shot isn’t as strongly considered in our paper. Please see item 1 in the general response – “Deep dive into zero-shot results for Forward Modeling”.
>
> Essentially, we found that sometimes zero-shot would produce estimates according to rational models, but when it did not it made the inexplicable decision to answer without considering the probabilities at all. Because of this, as well as zero-shot performing worse than CoT in general in our experiments, zero-shot responses would likely see limited use in practice. From our analysis, it was clear that CoT was doing something much more meaningful, and thus it was more reasonable to make a conclusion based on the CoT performance.

---

> ### Author Response · Authors · 2024-11-25
> **Response to Reviewer HdFJ part 2**
>
> ## Question 1:
> “How do you measure the correlation for task 2 and 3 in the forward modeling task?”
>
> For the analyses of the forward modeling task, we use Spearman correlation and Pearson correlation. We measure the correlations between the proportion of humans that chose one gamble over the other and the proportion of times the LLM predicts the human would choose one gamble over the other. This is computed across the 9,831 problem subset of the choices13k dataset, for which the data had around 15 human participants each. We compared correlations across human responses and LLM {zero-shot / CoT} responses, as well as between the maximum expected value responses and LLM {zero-shot / CoT responses}.
>
> For task 2, the model has only one output (the proportion of participants that would choose one gamble over the other), and this is used directly as the proportion prediction of the LLM and is compared with the proportion of human participants that actually chose that gamble.
>
> For task 3, the correlation computation is exactly the same as for task 1, where the model’s decisions are aggregated into a proportion, and this is then used to compute correlation against the real human data.
>
> We have updated the main text in blue to make this more clear.
>
> ## Question 2:
> “Could there be circular inconsistencies in the pairwise ranking for the inverse modeling task? For instance, if the model ranks decision A > B, B > C, but then C > A, how is this resolved?”
>
> Yes - there could be. Since the final output of the human experiment was an ordering of the decisions, we simply performed a count of the “pairwise wins” of each decision (ties were 0.5). Since we prompted for all pairs, counting directly corresponds to an aggregated ordering of the decisions.
>
> ## Question 3:
> “Why not use “average human participant” for the prompts in Table 5 in the Appendix?”
>
> Because we have more human data than just averages – we have individual responses. To effectively compare the LLM against the human data, we would also want the LLM to try to output the full distribution of human responses. These are used in the more advanced forward modeling analyses with the 18 decision models, highlighted in Appendix D. Simply having the averages would not enable us to perform these more detailed analyses.
>
> Your “average human participant” suggestion actually closely aligns with our task 2 in forward modeling, where the model is predicting the proportion of human participants that would select one gamble over the other.
>
> ## Question 4:
> “Why did you decide to use “you” and refer this to human simulation? “What would an (average) human participant do?” sounds better?”
>
> Using the second-person framing of “you” is common in the literature that investigates the use of LLMs to simulate human survey responses or psychology experiments (e.g., [1-3]). Furthermore, we wanted to sufficiently distinguish this from the prediction task (task 1), where the prompt is “A person is presented [...] Which machine does the person choose?”. Asking what a human participant would do isn’t much different.
>
> ## Question 5:
> In Table 11 in the Appendix, where is Bag 2 in Choice 2? Is it referring to the Bag 2 from above?
>
> No, there are a small subset of choices from the original psychology experiment [4] where there were no alternatives for the person to choose (see Table 12). We chose to replicate this in the LLM experiments because it enables us to perform more direct analyses between LLM and human responses. We specified this to the model in the prompt in Table 11: “For example, when there is only one bag, the person has no choice but to choose it.” We highlight this in blue for your convenience.
>
> [1] Lisa P Argyle, Ethan C Busby, Nancy Fulda, Joshua R Gubler, Christopher Rytting, and David Wingate. 2023. Out of one, many: Using language models to simulate human samples. Political Analysis 31, 3 (2023), 337–351.
>
> [2] Ryan Liu, Howard Yen, Raja Marjieh, Thomas L. Griffiths, and Ranjay Krishna. 2023. Improving interpersonal communication by simulating audiences with language models. arXiv:2311.00687
>
> [3] Perttu Hämäläinen, Mikke Tavast, and Anton Kunnari. 2023. Evaluating large language models in generating synthetic HCI research data: A case study. Proceedings of the 2023 CHI Conference on Human Factors in Computing Systems (2023).
>
> [4] Alan Jern, Christopher G Lucas, and Charles Kemp. 2017. People learn other people’s preferences through inverse decision-making. Cognition 168 (2017), 46–64.

---

> > ### Author Response · Authors · 2024-12-02
> > **Gentle Reminder for Response**
> >
> > Dear Reviewer HdFJ,
> >
> > We just wanted to provide a kind reminder that the discussion period is coming to a close in less than 24 hours. We would highly appreciate any remarks or comments in response to our rebuttal!
> >
> > Sincerely,
> >
> > Authors

---

> > > ### Comment · Reviewer_HdFJ · 2024-12-02
> > >
> > > Thank you for the detailed response and and taking extra effort for the additional experiments. I appreciate them. I have a follow up question, what model did you use for the experiments in the general response?

---

> ### Author Response · Authors · 2024-12-02
> **Response to Reviewer HdFJ**
>
> Thank you for your response, we greatly appreciate your question! All our additional experiments are conducted on the newest version of GPT-4o.
>
> Please also let us know if you think it makes sense to include any other models!
>
> Best,
> Authors

---

> > ### Comment · Reviewer_HdFJ · 2024-12-03
> >
> > Thank you. I believe incorporating at least one more model (e.g., Llama 3.1) would greatly benefit the readers. I strongly recommend the authors to include all of these experiments in the updated draft, as this would make the paper significantly more robust. I appreciate the authors' engagement and their extensive additional experiments. As a result, I am raising my score.

---

> ### Author Response · Authors · 2024-12-03
> **Response to Reviewer HdFJ**
>
> Thank you so much! We will be sure to add at least results for Llama 3.1 for our additional experiments, and we will try to also include other models such as Claude and Llama 3! We will write up all of the results in the final version --- we really appreciate your thorough engagement with our work throughout the review process, as well as your timely feedback and constructive opinions.
>
> Best,
>
> Authors

---

### Official Review · Reviewer_kdQx · 2024-11-04

**Soundness:** 3
**Presentation:** 3
**Contribution:** 2
**Rating:** 6
**Confidence:** 3

**Summary:**

Mainly the paper prompts LLMs (Llama3, Claude Opus, GPT-4) to predict the human choices in two games (predicting gambling choices and inferring preferences from choices). They find that LLMs poorly predict human behavior in both the zero-shot and COT settings—in the latter, the error is explained well by the models predicting the human behavior to be more rational than it is.

**Strengths:**

- applies research from the psychology field
- good discussion of alignment

**Weaknesses:**

see questions

**Questions:**

- Why set the temperature to 0.7 for some experiments?
- If I'm reading correctly, the poor predictions in the zero-shot setting aren't explained well by the model incorrectly modeling humans as rational actors? What's your alternative explanation?

> We found that GPT-4 Turbo could not provide a valid output ranking 47 choices at once. Thus, for all LLMs, we obtained rankings by asking the LLM to perform pairwise comparisons across 47choose2 pairs of decisions. Pairwise outputs were limited to {stronger, weaker, tie}, and were aggregated across decisions to form a ranking. Ties were discouraged to capture small differences between decisions

Why even permit the "tie" answer?

---

> ### Author Response · Authors · 2024-11-25
> **Response to Reviewer kdQx**
>
> Thank you so much for your review! We address each of your weaknesses / questions, please see below.
>
> ## Question 1:
> “Why set the temperature to 0.7 for some experiments?”
>
> Thank you for your question! We have rerun the models at a variety of different temperatures, and find results consistent with our original findings! Please check out item 2 in the general reviewer response.
>
> ## Question 2:
> “If I'm reading correctly, the poor predictions in the zero-shot setting aren't explained well by the model incorrectly modeling humans as rational actors? What's your alternative explanation?”
>
> Thank you for this feedback as well! From your feedback, we realized that our full analyses were not presented well here and have changed this! Please take a look at item 1 in the general reviewer response for our changes.
>
> ## Question 3:
> “Why even permit the "tie" answer?”
> For some decisions that were used in the inverse modeling psychology study, choices were not meaningfully different from each other. For instance, in the two decisions shown in the first and second rows of Table 12, the choices are not meaningfully different because in both cases the person only has one choice. If we did not enable the “tie” answer, we would just be injecting noise into the results.
>
> We actually debated whether to keep these decisions in our LLM study, but decided to keep them because it enabled us to more easily conduct comparisons between the existing human results and our LLM results.

---

> > ### Author Response · Authors · 2024-12-02
> > **Gentle Reminder for Response**
> >
> > Dear Reviewer kdQx,
> >
> > We just wanted to provide a kind reminder that the discussion period is coming to a close in less than 24 hours. We would highly appreciate any remarks or comments in response to our rebuttal!
> >
> > Sincerely,
> >
> > Authors

---

### Official Review · Reviewer_2ERM · 2024-11-06

**Soundness:** 3
**Presentation:** 3
**Contribution:** 3
**Rating:** 5
**Confidence:** 4

**Summary:**

This paper studies how advanced LLMs make predictions of human decisions, and reveal that frontier LMs assume that people are more rational than we really are. By using established datasets capturing human decisions, this study show that LLMs appear to be aligned with the human expectation that other people will act rationally (much like expected value theory), rather than with how people actually act.

**Strengths:**

- The application of psychometric data of human for analyzing LLMs behaviors reveals very interesting insights between the gap of LM vs. human decision-making.
- The study design and the analysis are rigorous.
- The insights of "While we have focused on using theories and paradigms from psychology to analyze LLMs, there may also be opportunities to use LLMs to refine existing theories about people." is thought-provoking.

**Weaknesses:**

- One might argue that these psychology tests used in this work appear to be quite simplistic to inform practical development of next generation LMs.

**Questions:**

- What are some other potential human decision-making models that can also be tested on LLMs, in addition to forward and inverse modeling?
- How are the insights gathered from this study inform the development of LMs applications, or inform better design of the next generation of LMs or other AIs?
- Would you imagine an ideal LMs in the future should align perfectly with humans on these psychology data? As we know that the conclusion of human data very much depend on who these people are. How do you imagine the different participant pool will impact the impact of the results of this study?

---

> ### Author Response · Authors · 2024-11-25
> **Response to Reviewer 2ERM**
>
> Thank you so much for your review! We are delighted that you found our paper to reveal very interesting insights, and our design and analysis to be rigorous!
>
> ## Weakness 1:
> “tests used in this work appear to be quite simplistic to inform practical development”
>
> The tests we used were derived from the literature on judgment and decision-making in psychology, where they are arguably *the* standard way to evaluate performance in these settings. Psychological experiments focus on simple stimuli to remove confounding factors. Since our experiments were based on comparisons to human data, we were limited in the tasks we could use.
>
> If we had found a positive result (such as models being able to accurately model humans), there would be more generalizability concerns. However, our findings are primarily negative and the takeaways are cautionary: That models may not have accurate representations of the decisions of their human partners, as demonstrated by the fact that they do not do so in the cases we consider. Under this lens, we believe the simplicity of the context is actually a strength of the paper, because it highlights that issues arise *even in the simplest settings.*
>
> We have also added to the paper some ways of introducing additional complexity into the tests. Item 3 in our general response contains an additional experiment evaluating LLM performance when the predicted people are of different demographic profiles, where we find that results are very similar across profiles.
>
> ## Question 1:
> “What are some other potential human decision-making models that can also be tested on LLMs, in addition to forward and inverse modeling?”
>
> Rather than being individual models, forward and inverse modeling are ultimately assumptions about the Bayesian nature of decision making. Forward models are normally “noisy” versions of a rational model, where options of greater utility are selected with higher probability (see Section 4, Rational models, Equation 1). Inverse models are obtained when we apply an additional Bayes’ rule (see Equation 2). Thus, forward and inverse models are actually a family of models that share the same set of very basic and intuitive assumptions that underlie the psychology literature.
>
> We do test many different models within the families of forward and inverse modeling. For instance, in the new paragraph in Section 3.1, we highlight our analyses conducted in Appendix D that involve 18 different forward models in order to explain the behavior of LLMs with zero-shot prompts. Similarly, in Section 4, “absolute utility”, “relative utility”, “likelihood”, and “marginal likelihood” are all different inverse models. Hopefully this helps clarify your concerns.
>
> ## Question 2:
> “How are the insights gathered from this study inform the development of LMs applications, or inform better design of the next generation of LMs or other AIs?”
>
> We see our results as identifying a significant concern in deploying LMs in settings where they will be asked to generate predictions about or simulate human behavior, or make decisions based on inferences from human choices. As indicated in the references provided in the paper, there are an increasing number of applications of this kind. So far, researchers have focused on cases where these models seem relatively successful in predicting, simulating, or making inferences from human behavior. Our work, through a detailed comparison against human choices across a large set of decision problems, highlights a setting where we find systematic deviations from how people make decisions.

---

> > ### Author Response · Authors · 2024-11-25
> > **Response to Reviewer 2ERM part 2**
> >
> > ## Question 3a:
> > “Would you imagine an ideal LMs in the future should align perfectly with humans on these psychology data?“
> > Generally, it is not necessarily the case that AI systems should completely replicate humans’ various psychological traits. However, we believe that specifically being able to accurately make predictions and inferences from people’s decisions is very desirable for LMs. The ability to predict and interpret people’s behavior is a precursor to identifying effective ways to provide assistance, simulating the helpfulness or harmlessness of a response, and learning individuals’ values and preferences, all of which are principal to the development of safe and beneficial AI systems.
> >
> > ## Question 3b:
> > “As we know that the conclusion of human data very much depend on who these people are. How do you imagine the different participant pool will impact the impact of the results of this study?”
> >
> > The forward and inverse modeling tasks we use have been employed in the most reputable studies in the field, were published in top-tier journals, and have laid the groundwork for an entire field of cognitive science, under which they have held up to numerous replication studies and follow-up works. Thus, we believe that the findings from these studies would not be affected based on the identities of the individuals involved.
> >
> > However, we believe an adjacent concern to yours is valid — that LLM outputs may be affected by the identities of the individuals. To address this, we conducted an additional experiment in item 3 of the general response targeting various human demographics and find that our results replicate across these identities.

---

> > > ### Comment · Reviewer_2ERM · 2024-11-25
> > >
> > > Thanks to the response! While I really appreciate the paper's effort in evaluating LM alignment to human decision-making using classical psychology resources, I found there lacks clear practical direct impact of these study results to what the authors identify as the successor applications---"identifying effective ways to provide assistance, simulating the helpfulness or harmlessness of a response, and learning individuals’ values and preferences, all of which are principal to the development of safe and beneficial AI systems." In addition, employing zero-shot prompts for representing forward and inverse modeling variations renders simplistic ways for applying LMs. Overall, despite the interesting problem setting, the technical contributions of this paper appear limited for ICLR.

---

> ### Author Response · Authors · 2024-12-04
> **Response to Reviewer 2ERM**
>
> Thank you for your continued engagement with our paper!
>
> We would like to quickly highlight why we believe each successor application is connected to our results for forward and inverse modeling:
>
> **Identifying effective ways to assist a person requires understanding what they want.** For instance, if a person is reaching out in the direction of an item, an embodied AI agent could take initiative and grab the item for the person. However, when the person's value function over items is incorrectly inferred by a robot (failure in inverse modeling), the AI agent might choose the wrong item to provide to the person. This could happen in an online setting, where a person looks for a long time at an advertisement but does not want the item, but an AI shopping assistant doesn't understand this and tries to help by adding the item to the person's cart.
>
> **Learning individuals’ values and preferences is a more generalized version of this**, where the model is learning the human's preferences, but the application might not be to provide assistance -- e.g., identifying malicious behavior.
>
> **Simulating the helpfulness and harmlessness of a response** is a technique where the language model has a list of potential utterances it can use to respond to a human, and chooses an utterance from this list by simulating the human's response to each potential utterance [1, 2], similar to conducting theory-of-mind. In this case, the model would specifically simulate the helpfulness and harmlessness of each response. In order to do so, the LLM would need to have an accurate forward model of how the human would react to and make decisions based on the utterance it provides, which is needed to estimate the outcomes and derive whether the statement was truly helpful/harmless. If the forward model is incorrect, it may lead the model to produce incorrect estimates of the human's decisions and thus incorrect estimates of the helpfulness/harmlessness of the response.
>
> We hope that this could help provide better intuition for why our experiments and findings matter in terms of these important applications in LLM use. We really appreciate your input, and **we will include a separate discussion section detailing these connections more deeply in the final version** to help clarify this for other readers.
>
> [1] Liu, R., Yen, H., Marjieh, R., Griffiths, T. L., & Krishna, R. (2023). Improving interpersonal communication by simulating audiences with language models. arXiv preprint arXiv:2311.00687.
>
> [2] Shaikh, O., Chai, V. E., Gelfand, M., Yang, D., & Bernstein, M. S. (2024, May). Rehearsal: Simulating conflict to teach conflict resolution. In Proceedings of the CHI Conference on Human Factors in Computing Systems (pp. 1-20).

---

### Author Response · Authors · 2024-11-24
**General Response**

## General Response [1/2]
We thank all the reviewers for their insightful comments!

We are glad that they found that our paper’s “topic is important” (dcdy), is “thought-provoking” (2ERM), and that it “invites follow-up work” (dcdy). We appreciate that they unanimously believe that we “link [our] work nicely to existing well-grounded cognitive science studies” (HdFJ), that our “setup is strongly grounded in well-known experiments” (dcdy), “applies research from the psychology field” (kdQx), and the “application of psychometric data [...] reveals very interesting insights” (2ERM).

We also appreciate that our reviewers found our work to have “very interesting direction and implication in the current LLM alignment research” (HdFJ), including a “good discussion of alignment (kdQx) and “important consequences for how trustworthy we deem their outputs” (dcdy).

We appreciate the positive feedback regarding our experiments, and are glad that reviewers have found our paper’s “design and analysis are rigorous” (2ERM), featuring a “wide variety of popular LLMs” leading to “insights in how the results generalize over different LLMs” (dcdy).

We also deeply appreciate their questions regarding generalizability across temperatures (kdQx, dcdy), participant groups / simulated human profiles (2ERM, HdFJ), and prompting strategies (dcdy). To address the reviewers’ concerns, we provide a set of **new experiments covering the issues that were raised**. We highlight the results for each experiment below.

We also appreciate the comments about lack of clarity in interpreting the zero-shot results (kdQx, HdFJ). The corresponding discussion (originally in the 2nd paragraph of 3.1 and Appendix D) was not properly presented in the main text, and we have **revised this section to elucidate our findings**. We have added an explanation below.

For reviewer-specific questions on evaluation decision details (e.g., measuring correlation from HdFJ), please refer to review-specific responses!

## 1. Deep dive into zero-shot results for Forward Modeling
When we found that models’ forward modeling zero-shot results were not well explained by maximum expected value, we conducted a fit across a wider range of 18 behavioral models to try to understand this behavior.

Specifically, we replicated the original analyses conducted by the authors of the choices13k dataset [1], which included
Heuristic models [2], where people are thought to use mental shortcuts to make decisions
Counterfactual models [2], which appeal to constructs like regret and disappointment
Subjective Expected Utility models, which assume that quantities involved (money and probability) are perceived or treated subjectively. This includes many of the most influential models including Expected Utility Theory, Prospect Theory [3], and the model proposed by [1], the Mixture of Theories (MOT).

In Appendix D, we provide the full analysis of the results of fitting these various models. The best interpretation we found uses MOT, where the model fit a mixture of two probability weighting functions — one which is linear (matching humans), and one that approximated a flat line. This suggests that the LLM often expected people to be rational (ie. the linear function), but also completely ignores all probabilities (i.e., weights them equally) a significant proportion of the time.

We have rewritten Section 3.1 to better highlight this result. This is in blue in the **updated PDF**. We thank the reviewers for pointing this out for us.

[1] Joshua C. Peterson, David D. Bourgin, Mayank Agrawal, Daniel Reichman, and Thomas L. Griffiths. 2021. Using large-scale experiments and machine learning to discover theories of human decision-making. Science 372, 6547 (2021), 1209–1214.
[2] Lisheng He, Wenjia Joyce Zhao, and Sudeep Bhatia. 2022. An ontology of decision models.
Psychological Review 129, 1 (2022), 49–72.
[3] Daniel Kahneman and Amos Tversky. 1979. Prospect theory: An analysis of decision under risk. Econometrica 47, 2 (1979), 263–292.

---

### Author Response · Authors · 2024-11-25
**General Response 2/2**

## 2. Additional experiment: Temperature
Originally, we chose temperature = 1 for some tasks and temperature = 0.7 for others, which the reviewers point out is inconsistent. 0.7 is another common temperature setting, used by works such as AlpacaEval [4] on instruction following tasks.

To further validate the robustness of our findings, we test the forward modeling experiment using temperatures 0.0, 0.5, 1.0, 1.5, and 2.0 for both zero-shot and CoT conditions, and compute spearman correlation with both human and max expected value, shown in the table below. We find that varying temperature does not substantially alter our findings other than CoT with high temperatures, where the model becomes unable to complete long responses due to noisy outputs.

| Zero-shot vs.|Max Expected Value | CoT vs. Max |Expected Value |
|----------|----------|----------|-----|
| Temp | Spearman | Temp | Spearman |
| 0.0  | 0.313 | 0.0   | 0.540 |
| 0.5  | 0.272 | 0.5   | 0.532 |
| 1.0  | 0.284 | 1.0   | 0.539 |
| 1.5  | 0.302 | 1.5   | 0.351 |
| 2.0  | 0.308 | 2.0   | model unable |

| Zero-shot vs.|Human | CoT vs. |Human|
|----------|----------|----------|-----|
| Temp | Spearman | Temp | Spearman |
| 0.0  | 0.533 | 0.0   | 0.540 |
| 0.5  | 0.381 | 0.5   | 0.477 |
| 1.0  | 0.341 | 1.0   | 0.518 |
| 1.5  | 0.489 | 1.5   | 0.528 |
| 2.0  | 0.389 | 2.0   | model unable |

[4] Li, X., Zhang, T., Dubois, Y., Taori, R., Gulrajani, I., Guestrin, C., Liang, P., & Hashimoto, T. B. (2023). AlpacaEval: An Automatic Evaluator of Instruction-following Models. GitHub repository. url: https://github.com/tatsu-lab/alpaca_eval

## 3. Additional experiment: Participant Groups and Prompting Variations
To alleviate some concerns about the simplicity of the experimental setup and lack of representation across demographic groups, we conducted a follow-up experiment replicating our original analysis on various demographic profiles varying age and gender. We find that our analyses stay consistent across all these factors, suggesting that our findings are at least somewhat robust to variations in context as well as differing profiles of people.

For additional prompt variation, we also prompt the LLM to make the same predictions for a monkey rather than a human. While we do not see differences in zero-shot results, when performing CoT models accurately reflect that monkeys are less rational and the resulting predictions are less correlated with max expected value. However, the correlations are also increased with actual human behavior, supporting our hypothesis with an example where the model predicts less rational decisions, but these are actually more in-line with how humans actually behave.

**Demographic + Prompt Variation Results**
| Zero-shot vs. |Max EV | |CoT vs. Max EV | | |
|----------|----------|----------|-----|-----|----|
| Demographic | Spearman | p-value    | Demographic | Spearman | p-value |
| Woman 18-35 | 0.289        | 9.44E-21 | Woman 18-35 | 0.970        | <1E-150 |
| Woman 45-60 | 0.313        | 4.23E-24 | Woman 45-60 | 0.972        | <1E-150 |
| Woman 65-85 | 0.289        | 9.44E-21 | Woman 65-85 | 0.969        | <1E-150 |
| Man 18-35      | 0.335        | 1.37E-27 | Man 18-35      | 0.971        | <1E-150 |
| Man 45-60      | 0.327        | 2.50E-26 | Man 45-60      | 0.965        | <1E-150 |
| Man 65-85      | 0.348        | 9.19E-30 | Man 65-85      | 0.972        | <1E-150 |
| Monkey           | 0.348        | 8.41E-30 | Monkey          | 0.654        | 7.37E-123 |

| Zero-shot vs. |Human | |CoT vs. Human | | |
|----------|----------|----------|-----|-----|----|
| Demographic | Spearman | p-value    | Demographic | Spearman | p-value |
| Woman 18-35 | 0.529        | 2.75E-73 | Woman 18-35 | 0.549        | 9.23E-80 |
| Woman 45-60 | 0.519        | 6.07E-70 | Woman 45-60 | 0.546        | 8.54E-79 |
| Woman 65-85 | 0.529        | 2.75E-73 | Woman 65-85 | 0.550        | 3.70E-80 |
| Man 18-35      | 0.499        | 4.18E-64 | Man 18-35      | 0.545        | 2.10E-78 |
| Man 45-60      | 0.530        | 1.43E-73 | Man 45-60      | 0.539        | 1.34E-76 |
| Man 65-85      | 0.508        | 8.54E-67 | Man 65-85      | 0.541        | 3.10E-77 |
| Monkey           | 0.482        | 2.18E-59 | Monkey          | 0.630        | 8.47E-112 |

---

### Author Response · Authors · 2024-12-04
**Discussion Period Summary**

To conclude the discussion period, we would like to provide a quick summary comprising of the discussions:

### 1. Better explanation for zero-shot case in forward modeling

Initially, reviewers kdQx and HdFJ expressed concerns about how to interpret the zero-shot results for the forward modeling. We have since updated the writing to provide readers a better understanding of this (see Section 3.1 blue text in PDF).

### 2. Strong results on additional experiments testing generalizability: {temperature, demographic prompts, prompting strategies}

In the original reviews, reviewers had concerns about generalizability across temperatures (kdQx, dcdy), participant groups / simulated human profiles (2ERM, HdFJ), and prompting strategies (dcdy). We conducted additional experiments on temperature and demographics (highlighted in the general response) that show that our findings are consistent across these attributes, which reviewers HdFJ and dcdy have since acknowledged:

"I appreciate the authors' engagement and their extensive additional experiments. As a result, I am raising my score." (HdFJ)

"Thank you for answering my questions and providing these additional experiments. Would be great to see those in the CR as well." (dcdy)

Both these reviewers have expressed the desire to see these additional experiments included in the main text, and reviewer HdFJ has requested additional models for these experiments, which we will happily comply for the camera ready version if the paper is accepted.

### 3. Discussion with reviewer 2ERM on practical impact on informing the development of LMs

Reviewer 2ERM brought up that 'there lacks clear practical direct impact of these study results to what the authors identify as the successor applications---"identifying effective ways to provide assistance, simulating the helpfulness or harmlessness of a response, and learning individuals’ values and preferences"'. We have provided an explanation on why we believe these cases are well-motivated, and plan to add this as a discussion section in the final version of the paper if it is accepted to help readers better understand the grounded impacts of this work.

Separately, it is worth mentioning that Reviewer kdQx has not responded to our rebuttal, so they were not able to update their opinion given our points 1 and 2 --- which they request in their original review.

### Conclusion
We thank our reviewers for engaging with our paper and providing invaluable feedback in their reviews and during the discussion period, and as our reviewers have found our work to have impacts in impacts towards alignment, strong connections to psychology literature, that the "topic is important" (dcdy) and the paper "reveals very important insights" (2ERM), we hope to be able to share our findings with a larger audience.

---

### Meta-Review · Program_Chairs · 2024-12-24

**Metareview:**

PC is entering meta-review on behalf of SAC/AC:

The reviewers felt that the paper was an interesting research direction, and had important implication in LLM alignment research demonstrating the limitations of models’ ability to understand genuine human behaviour.

**Additional Comments On Reviewer Discussion:**

TBD

---

### Decision · Program_Chairs · 2025-01-22

Accept (Poster)